# Decomposing The Dark Matter of Sparse Autoencoders

**Joshua Engels**                                                                                 *jengels@mit.edu*
*MIT*

**Logan Smith**                                                                         *logansmith5@gmail.com*
*Independent*

**Max Tegmark**                                                                                 *tegmark@mit.edu*
*MIT & IAIFI*

**Reviewed on OpenReview:** *https://openreview.net/forum?id=sXq3Wb3vef*

## Abstract

Sparse autoencoders (SAEs) are a promising technique for decomposing language model activations into interpretable linear features. However, current SAEs fall short of completely explaining model performance, resulting in "dark matter": unexplained variance in activations. This work investigates dark matter as an object of study in its own right. Surprisingly, we find that much of SAE dark matter—about half of the error vector itself and $> 90\%$ of its norm—can be linearly predicted from the initial activation vector. Additionally, we find that the scaling behavior of SAE error norms at a per token level is remarkably predictable: larger SAEs mostly struggle to reconstruct the same contexts as smaller SAEs. We build on the linear representation hypothesis to propose models of activations that might lead to these observations. These insights imply that the part of the SAE error vector that cannot be linearly predicted ("nonlinear" error) might be fundamentally different from the linearly predictable component. To validate this hypothesis, we empirically analyze nonlinear SAE error and show that 1) it contains fewer not yet learned features, 2) SAEs trained on it are quantitatively worse, and 3) it is responsible for a proportional amount of the downstream increase in cross entropy loss when SAE activations are inserted into the model. Finally, we examine two methods to reduce nonlinear SAE error: inference time gradient pursuit, which leads to a very slight decrease in nonlinear error, and linear transformations from earlier layer SAE outputs, which leads to a larger reduction.

## 1 Introduction

The ultimate goal for ambitious mechanistic interpretability is to understand neural networks from the bottom up by breaking them down into programs ("circuits") and the variables ("features") that those programs operate on (Olah, 2023). One recent successful technique for finding features in language models has been sparse autoencoders (SAEs), which learn a dictionary of one-dimensional representations that can be sparsely combined to reconstruct model hidden activations (Cunningham et al., 2023; Bricken et al., 2023). However, as observed by Gao et al. (2024), the scaling behavior of SAE width (number of latents) vs. reconstruction mean squared error (MSE) is best fit by a power law with a constant error term. Gao et al. (2024) speculate that this component of SAE error below the asymptote might best be explained by model activations having components with denser structure than simple SAE features (e.g. Gaussian noise). This is a concern for the ambitious agenda because it implies that there are components of model hidden states that are harder for SAEs to learn and which might not be eliminated by simple scaling of SAEs.

Motivated by this discovery, in this work our goal is to specifically study the SAE error vector itself, and in doing so gain insight into the failures of current SAEs, the dynamics of SAE scaling, and possible distributions of model activations. Thus, our direction differs from the bulk of prior work that seeks to quantify SAE

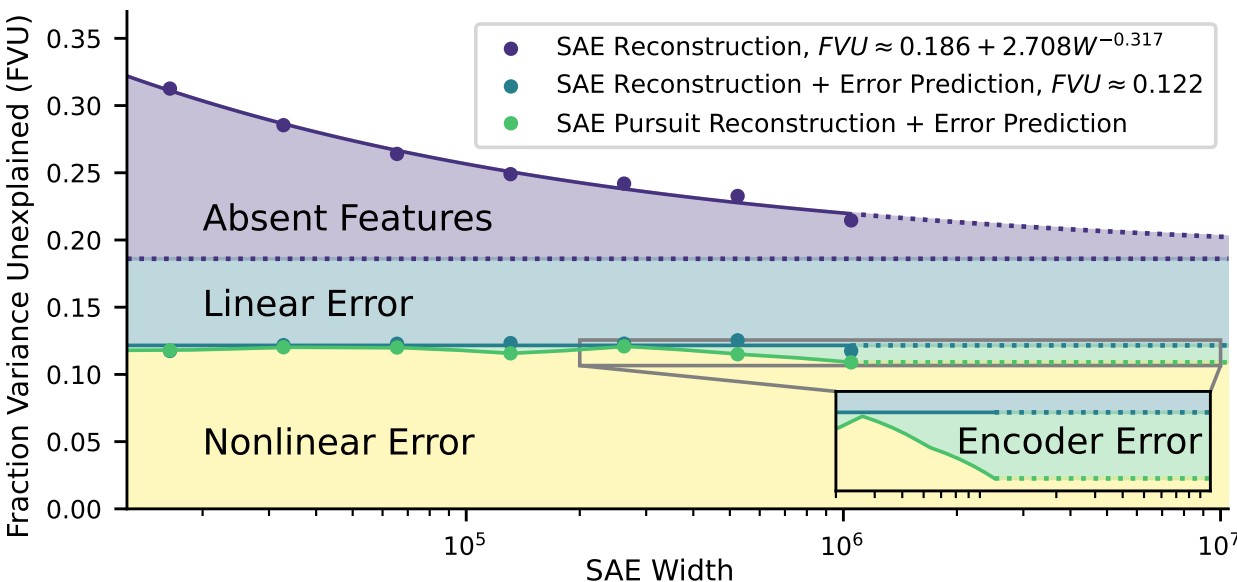

Figure 1: A breakdown of SAE dark matter for layer 20 Gemma 9B SAEs, with dotted lines assuming that observed trends continue for larger SAEs. See Section 4 for how we break down the overall fraction of unexplained variance into absent features, linear error, and nonlinear error. See Section 6.1 for further separating encoder error from nonlinear error.

failures, as these mostly focus on downstream benchmarks or simple cross entropy loss (see e.g. Gao et al. (2024); Templeton et al. (2024); Anders & Bloom (2024)). The structure of this paper is as follows:

1. In Section 4, we introduce the fundamental mystery that we will explore throughout the rest of the paper: SAE errors are shockingly predictable. To the best of our knowledge, we are the first to show that a large fraction of SAE error vectors can be explained with a linear transformation of the input activation, that the norm of SAE error vectors can be accurately predicted by a linear projection of the input activation, and that on a per-token level, error norms of large SAEs are linearly predictable from small SAEs.

2. In Section 5, we investigate the linearly predictable and nonlinearly predictable components of SAE error. We find that although the nonlinear component affects downstream cross entropy loss in proportion to its norm, it is qualitatively different from linear error: as compared to linear error, nonlinear error is harder to learn SAEs for and has a norm that is harder to linearly predict from activations, suggesting that it consists of a smaller proportion of not-yet-learned linear features.

3. In Section 6, we investigate methods for reducing nonlinear error. We show that inference time optimization increases the fraction of variance explained by SAEs, but only slightly decreases nonlinear error. Additionally, we show that we can use SAEs trained on previous components to decrease nonlinear error and total SAE error.

## 2   Related Work

**Language Model Representation Structure:** The linear representation hypothesis (LRH) (Park et al., 2023; Elhage et al., 2022) claims that language model hidden states can be decomposed into a sparse sum of linear feature directions. The LRH has seen recent empirical support with *sparse autoencoders* (Makhzani & Frey, 2013; Bricken et al., 2023; Cunningham et al., 2023), which have succeeded in decomposing much of the variance of language model hidden states into such a sparse sum, as well as a long line of work that has used probing and dimensionality reduction to find causal linear representations for specific concepts (Alain, 2016; Nanda et al., 2023; Marks et al., 2024; Gurnee, 2024). On the other hand, some recent work has questioned whether the linear representation hypothesis is true: Engels et al. (2024) find multidimensional

circular representations in Mistral (Jiang et al., 2023) and Llama (AI@Meta, 2024), and Csordás et al. (2024) examine synthetic recurrent neural networks and find "onion-like" non-linear features not contained in a linear subspace. This has inspired recent discussion about what a true model of activation space might be: Mendel (2024) argues that the linear representation hypothesis ignores the growing body of results showing the multi-dimensional structure of SAE latents, and Smith (2024b) argues that we only have evidence for a "weak" form of the superposition hypothesis holding that only *some* features are linearly represented (such a hypothesis has also been studied in the sparse coding literature (Tasissa et al., 2020)).

**SAE Errors and Benchmarking:** Multiple works have introduced techniques to benchmark SAEs and characterize their error: Bricken et al. (2023), Gao et al. (2024), and Templeton et al. (2024) use manual human analysis of features, automated interpretability, downstream cross entropy loss when SAE reconstructions are inserted back into the model, and feature geometry visualizations; Karvonen et al. (2024) use the setting of board games, where the ground truth features are known, to determine what proportion of the true features SAEs learn; and Anders & Bloom (2024) use the performance of the model on NLP benchmarks when the SAE reconstruction is inserted back into the model. More specifically relevant to our main direction in this paper, Gurnee (2024) finds that SAE reconstruction errors are *pathological*, that is, when SAE reconstructions are inserted into the model, they have a larger effect on cross entropy loss than random perturbations with the same error norm. Follow up work by Heimersheim & Mendel (2024) and Lee & Heimersheim (2024) find that this effect disappears when the random baseline is replaced by a perturbation in the direction of the difference between two random activations.

**SAE Scaling Laws:** Anthropic (2024), Templeton et al. (2024), and Gao et al. (2024) study how SAE MSE scales with respect to FLOPS, sparsity, and SAE width, and define scaling laws with respect to these quantities. Templeton et al. (2024) also study how specific groups of language features like chemical elements, cities, animals, and foods are learned by SAEs, and show that SAEs predictably learn these features in terms of their occurrence. Finally, Bussmann et al. (2024) find that larger SAEs learn two new types of dictionary vectors as comparsed to smaller SAEs: features not present at all in smaller SAEs, and more fine-grained "feature split" versions of features in smaller SAEs.

## 3 Notation

We consider neural network activations $\mathbf{x} \in \mathbb{R}^d$ and sparse autoencoders $\texttt{Sae} \in \mathbb{R}^d \to \mathbb{R}^d$. Sparse autoencoders map inputs to a latent space $\mathbb{R}^m$ with $m \gg d$ and then back into $\mathbb{R}^d$, while also requiring that only a sparse set of the $m$ latent dimensions are nonzero. SAEs have the general architecture

$$\texttt{hidden}(\mathbf{x}) \coloneqq \sigma(\boldsymbol{W}_{enc} \cdot (\mathbf{x} - \boldsymbol{b}_{dec}) + \boldsymbol{b}_{enc}) \tag{1}$$

$$\texttt{Sae}(\mathbf{x}) \coloneqq \boldsymbol{W}_{dec} \cdot \texttt{hidden}(x) + \boldsymbol{b}_{dec} \tag{2}$$

where $\sigma$ is an architecture dependent activation function, and seek to minimize

$$\mathcal{L}_{\texttt{Sae}(\mathbf{x})} = \|\mathbf{x} - \texttt{Sae}(\mathbf{x})\|_2^2 + S(\texttt{hidden}(\mathbf{x})) \tag{3}$$

where $S$ is an architecture dependent sparsity function. For convenience, we define $\texttt{SaeError}(\mathbf{x})$ as the reconstruction error of the SAE and $L_0$ as the average number of nonzeros in $\texttt{hidden}(\mathbf{x})$:

$$\texttt{SaeError}(\mathbf{x}) \coloneqq \mathbf{x} - \texttt{Sae}(\mathbf{x}) \tag{4}$$

$$L_0 \coloneqq E_{\mathbf{x}}(\|\texttt{hidden}(\mathbf{x})\|_0) \tag{5}$$

In this work, we study the relationship between $\mathbf{x}$ and $\texttt{SaeError}(\mathbf{x})$ and are agnostic to the specifics of SAE architecture. Thus, we examine both TopK SAEs (Gao et al., 2024) and JumpRelu SAEs (Rajamanoharan et al., 2024). The TopK SAE is defined as

$$\sigma_{TopK}(\mathbf{z}) = \text{set all but the } k \text{ largest dimensions of } \mathbf{z} \text{ to zero} \tag{6}$$

$$S_{TopK}(\texttt{hidden}(\mathbf{x})) = 0 \tag{7}$$

Note that $L_0 = k$ for a TopK SAE. The JumpRelu SAE is defined as

$$\sigma_{JumpRelu}(\mathbf{z})_i = \begin{cases} \mathbf{z}_i & \text{if } \mathbf{z}_i - \theta_i > 0 \\ 0 & \text{otherwise} \end{cases} \tag{8}$$

$$S_{JumpRelu}(\texttt{hidden}(\mathbf{x})) = \lambda \left\| \texttt{hidden}(\mathbf{x}) \right\|_0 \tag{9}$$

where $\theta$ is a learned bias vector and $\lambda$ is a sparsity parameter.

## 4 Predicting SAE Error

In this section, we evaluate the extent to which SAE errors $\texttt{SaeError}(\mathbf{x})$ can be predicted from input model activations $\mathbf{x}$.

We run experiments [1] on Gemma 2 2B and 9B (Team et al., 2024) and Llama 3.1 8B (AI@Meta, 2024). We use the suite of Gemma Scope (Lieberum et al., 2024) sparse autoencoders for Gemma 2 2B experiments and Llama Scope (He et al., 2024) for Llama 3.1 8B. For experiments where we analyze the effect of $L_0$ and SAE width, we focus on layer 12 of Gemma 2 2B and layer 20 of Gemma 9 9B in the main text as these layers have the most SAEs in Gemma Scope, and we include additional layers in the appendix (we do not run on Llama Scope for these experiments, as Llama Scope has just one SAE $L_0$ and two SAE widths per layer). For experiments where we analyze the effect of layer, we show the set of SAEs with $L_0$ closest to a target $L_0$ across all layers for both Gemma Scope and Llama Scope suites of SAEs.

We use 300 contexts of 1024 tokens from the uncopywrited subset of the Pile (Gao et al., 2020) and then filter to only activations of tokens after position 200 in each context, as Lieberum et al. (2024) find that earlier tokens are easier for sparse autoencoders to reconstruct, and we wish to ignore the effect of token position on our results. This results in a dataset of about 247k activations. For linear regressions, we use a random subset of size 150k as training examples (since all models have a dimension of less than 5000, this prevents overfitting) and report the $R^2$ on the other 97k activations. For linear transformations to a multi-dimensional output, we report the average $R^2$ across dimensions. We include bias terms in our linear regressions but omit them from equations for simplicity.

### 4.1 Predicting SAE Error Norm

For our first set of experiments, we find the optimal linear probe $\boldsymbol{a}^*$ from $\mathbf{x}$ to $\|\texttt{SaeError}(\mathbf{x})\|_2^2$. Formally (with a slight abuse of notation, since $\mathbf{x}$ is a random variable and not a dataset), we solve for

$$\boldsymbol{a}^* = \arg\min_{\boldsymbol{a} \in \mathbb{R}} \left\| \boldsymbol{a}^T \cdot \mathbf{x} - \|\texttt{SaeError}(\mathbf{x})\|_2^2 \right\|_2 \tag{10}$$

The $R^2$ of these probes are all extremely high: across all combinations of SAE width, $L_0$, layer, and model, between 70% and 95% of the variance in SAE error norm is explained by the optimal linear probe.

**Results across SAE width and $L_0$:** We plot the $R^2$ of probes from Eq. (10) across SAE width and SAE $L_0$ for layer 9 of Gemma 2 2B and layer 20 of Gemma 2 9B as a contour plot (on the left) in Fig. 2. We also include plots for additional layers in Appendix A. Overall, sparser and wider SAEs have less predictable error norms, although interestingly for some layers, extremely dense SAEs have less predictable error norm as well.

**Results across layers and models:** In Fig. 3 we plot the $R^2$ of probes from Eq. (10) for the SAEs with $L_0$ closest to 50 on Gemma 2 2B and 9B and Llama Scope SAEs (which all have $k = L_0 = 50$). We find that the $R^2$ is low for the first few layers, increases rapidly to a value of about 0.8 to 0.95, remains at this level with a slight dip for mid-late layers, and then drops off steeply in the last few layers. In Appendix A, we compare the $R^2$ of this approach across layers versus other baselines like token identity; we find that using activations does much better, and thus it is not "easy" to predict error norm.

---

[1] Code at `https://anonymous.4open.science/r/SAE-Dark-Matter-1163`

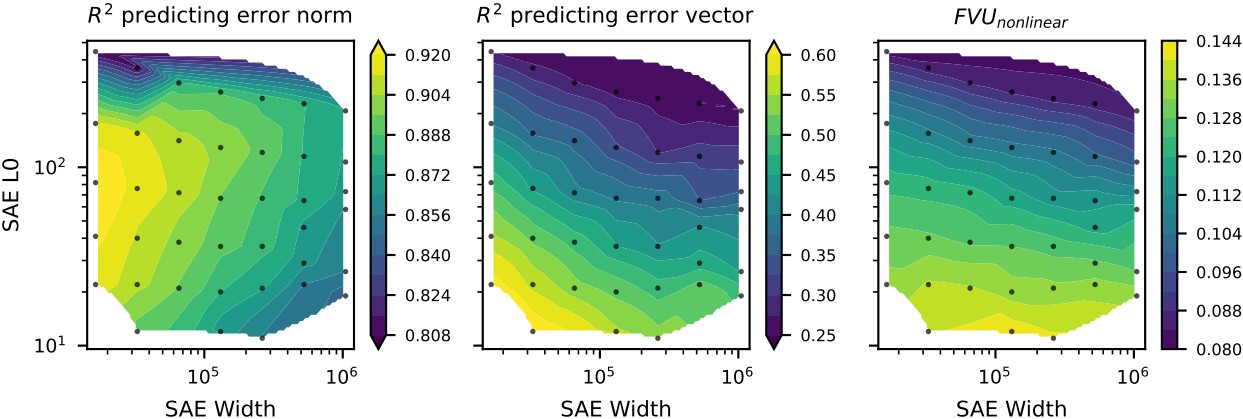

(a) Linear prediction results for layer 12 Gemma 2 2B SAEs from Gemma Scope. FVU_nonlinear is roughly constant for a fixed $L_0$.

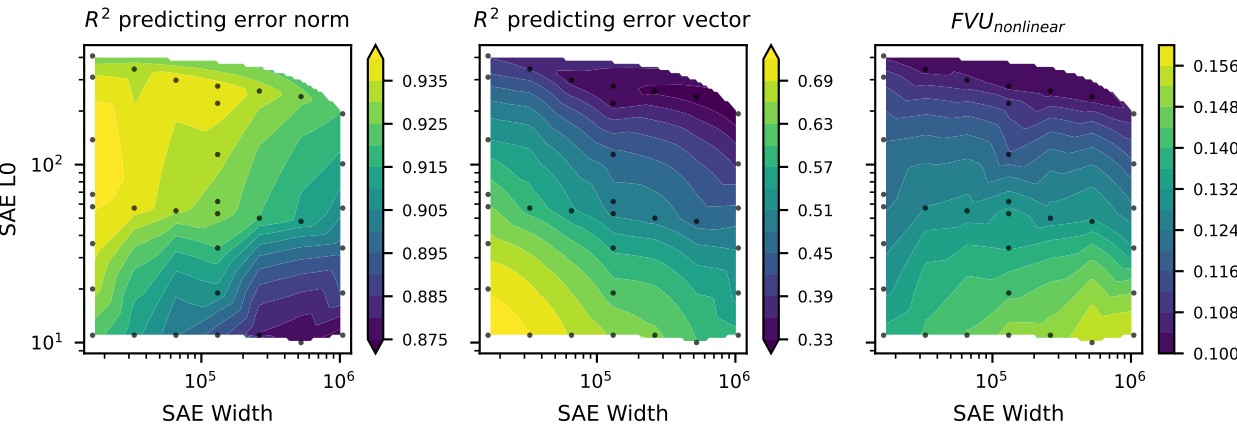

(b) Linear prediction results for layer 20 Gemma 2 9B SAEs from Gemma Scope. FVU_nonlinear is roughly constant for a fixed $L_0$.

Figure 2: Results of linearly predicting SAE error norm and SAE error from model activations on Gemma 2 2B layer 12 (**top**) and Gemma 2 9B layer 20 (**bottom**). The right plots show the $R^2$ of predicting SAE error norms (see Eq. (10)), the middle plots show the $R^2$ of predicting SAE error vectors (see Eq. (11), and the right plots show $1 - R^2$ of predicting model activations given the SAE reconstruction and the SAE error vector prediction. We note that these 2D heatmaps are somewhat sparse and only the black dots represent actual SAEs. This is because we use Gemma Scope SAEs, which are trained only on some $L_0$s and SAE widths. We do a linear interpolation between SAEs to predict $R^2$ between hyperparameters.

## 4.2 Predicting SAE Error Vectors

We next examine the $R^2$ of the optimal linear transform $\boldsymbol{b}^*$ from $\mathbf{x}$ to SaeError($\mathbf{x}$):

$$\boldsymbol{b}^* := \underset{\boldsymbol{b}\in\mathbb{R}^{d\times d}}{\arg\min} \|\boldsymbol{b} \cdot \mathbf{x} - \texttt{SaeError}(\mathbf{x})\|_2 \tag{11}$$

As we show in Fig. 2 (middle column) and in Appendix A for additional layers, the $R^2$ of these transforms varies widely, between 15% and 72%; this is less than the $R^2$ for our norm prediction experiments, but still much higher than we might expect. Intuitively, this result implies that there are large linear subspaces that the SAE is mostly failing to learn. There is also a clear pattern across SAE $L_0$ and width: $R^2$ decreases with increasing SAE width and $L_0$. Interestingly, this pattern is *not* the same as it was above for SAE error norm: the $R^2$ of error norm predictions increases with SAE $L_0$, while it decreases for error vector predictions. One

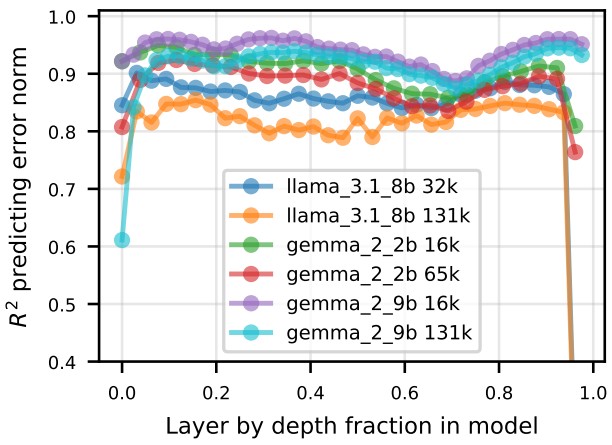
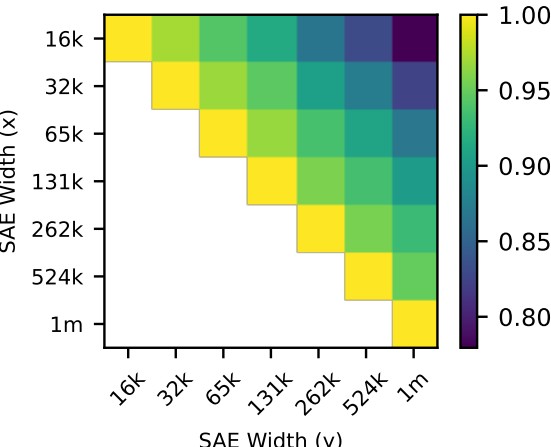

Figure 3: $R^2$ of SAE error norm predictions (see Eq. (10)) for Gemma Scope SAEs of width $L_0 \approx 50$ and Llama Scope SAEs of width $L_0 = k = 50$.

Figure 4: $R^2$ for linear probes of per token SAE errors of larger SAEs from smaller SAEs. Prediction accuracy decreases as the SAEs get farther apart in scale, but overall remains high.

concern might be that $\boldsymbol{b}^*$ is mostly reversing feature shrinkage; in Appendix A.1, we show that this is not the case.

## 4.3 Nonlinear FVU and Breaking Down SAE Dark Matter

Another related metric we are interested in is the total amount of the original activation $\mathbf{x}$ we fail to explain using *both* $\texttt{Sae}(\mathbf{x})$ and a linear projection of $\mathbf{x}$. That is, assuming we have found $\boldsymbol{b}^*$ as in Eq. (11), we are interested in the fraction of variance unexplained (FVU) by the sum of $\texttt{Sae}(\mathbf{x})$ and $\boldsymbol{b}^* \cdot \mathbf{x}$:

$$\texttt{FVU}_{\texttt{nonlinear}} := 1 - R^2(\mathbf{x}, \texttt{Sae}(\mathbf{x}) + \boldsymbol{b}^* \cdot \mathbf{x}) \tag{12}$$

We label this quantity $\texttt{FVU}_{\texttt{nonlinear}}$ because it is intuitively the amount of the SAE's unexplained variance that is not a linear projection of the input. Interestingly, we find that for middle layers (layer 12 Gemma 2 2B and layer 20 Gemma 2 9B) at a fixed $L_0$, $\texttt{FVU}_{\texttt{nonlinear}}$ is approximately constant (see Fig. 2), while for other layers it decreases slightly as $L_0$ increases (see Appendix A). That is, even though we can linearly predict a smaller portion of the error vector in larger SAEs, this effect is counteracted by the fact that the SAE error vector itself is getting smaller. In contrast, $\texttt{FVU}_{\texttt{nonlinear}}$ decreases as SAE $L_0$ increases.

As stated above, for middle layers we observe that $\texttt{FVU}_{\texttt{nonlinear}}$ is roughly constant at a fixed sparsity. Because we observe this across multiple orders of magnitude of SAE scale, we hypothesize that this will continue to hold as we scale the SAE. Using this assumption, we plot a horizontal fit for $\texttt{FVU}_{\texttt{nonlinear}}$ and a power law fit with a constant for Gemma SAE reconstructions as SAE width scales (choosing the SAEs closest to $L_0 \approx 60$) in Fig. 1. The power law fit asymptotes above the horizontal fit, which implies the presence of linear error even at very large SAE width. The absent features component of the error in Fig. 1 comes from the following observation: if our hypothesis is correct that nonlinear error is roughly constant as the layer 20 SAEs scale, then the nonlinear features must reside in the linear error component, since this is the only component that decreases as the SAE scales and learns more features. We investigate this hypothesis further in Appendix B.2.

## 4.4 Predicting SAE Per-Token Error Norms

We now examine per-token SAE scaling behavior. Given two SAEs, $\texttt{SAE}_1$ and $\texttt{SAE}_2$, we are interested in how much of the variance in error norms in $\texttt{SAE}_2$ is predictable from error norms in $\texttt{SAE}_1$. That is, we want to

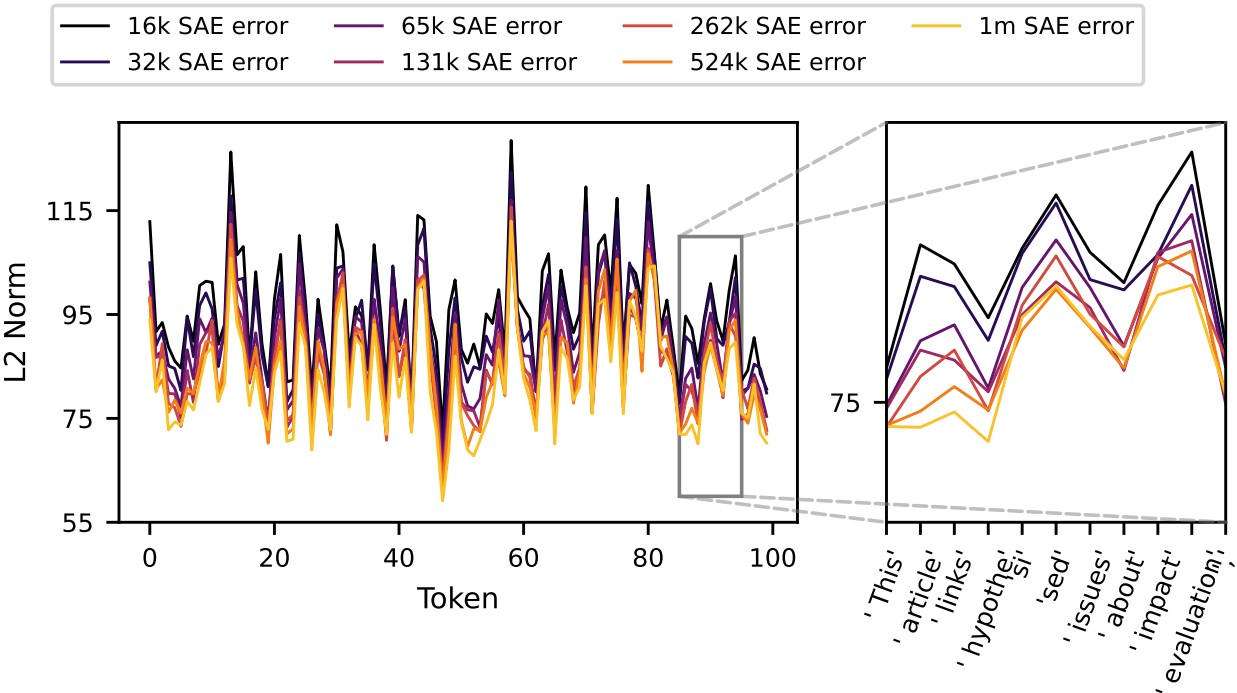

Figure 5: Per token scaling with average nonlinear error, layer 20 Gemma 9B SAEs from Gemma Scope closest to $L_0 = 60$.

find $\boldsymbol{c}^*$ such that

$$\boldsymbol{c}^* := \underset{\boldsymbol{c} \in \mathbb{R}}{\arg\min} \left\| \boldsymbol{c} \cdot \left\| \texttt{SaeError}_1(\mathbf{x}) \right\|_2^2 - \left\| \texttt{SaeError}_2(\mathbf{x}) \right\|_2^2 \right\|_2 \tag{13}$$

Note that in practice, although $\boldsymbol{a}^*$ also can predict the norm of SAE error, it requires training the target SAE to learn a probe. Here, on the other hand, although we formulate finding $\boldsymbol{c}^*$ as an optimization problem that requires a larger SAE, in practice we do not need to actually train the larger SAE to get interesting insights: since $\boldsymbol{c}^*$ has just one component, it simply measures how well small SAE error can be multiplied by a scalar to predict large SAE error. If the $R^2$ is high, we know that on tokens that small SAEs perform poorly on, larger SAEs will as well. In Fig. 4, we plot the $R^2$ of $\boldsymbol{c}^*$ probes on all pairs of Gemma Scope 9B layer 20 SAEs with $L_0 \approx 60$ (restricting to pairs where $\texttt{SAE}_2$ is larger than $\texttt{SAE}_2$), and find that indeed, per token SAE errors are highly predictable. Additionally, we show concretely what these correlated SAE error norms looks like on a set of 100 tokens from the Pile in Fig. 5.

## 5 Analyzing Components of SAE Error

In previous sections, we broke down the SAE error vector into a linearly predictable component and a non-linearly predictable component. A reasonable question is whether these error subsets meaningfully differ. Thus, in this section, we run experiments on these components to determine how the linear and non-linear components of $\texttt{SaeError}(\mathbf{x})$ differ. For convenience, given a probe $\boldsymbol{b}^*$ from Eq. (11), we write

$$\texttt{LinearError}(\mathbf{x}) := \boldsymbol{b}^* \cdot \mathbf{x} \tag{14}$$

$$\texttt{NonlinearError}(\mathbf{x}) := \texttt{SaeError}(\mathbf{x}) - \texttt{LinearError}(\mathbf{x}) = \texttt{SaeError}(\mathbf{x}) - \boldsymbol{b}^* \tag{15}$$

In Appendix B.2 we discuss a hypothesis for SAE feature activations that might explain *why* we observe these differences; this appendix section is not necessary for understanding the experiments in this section, and we will refer to it in the few cases where it provides additional intuition.

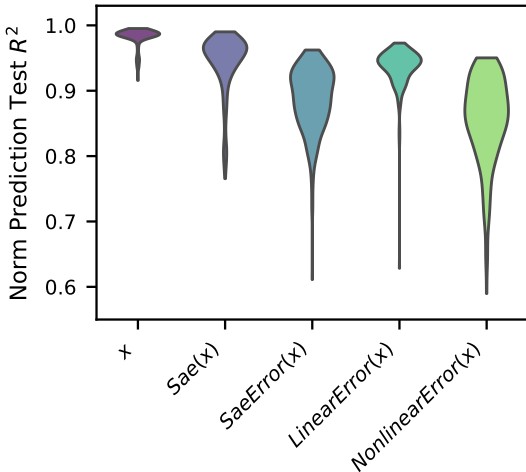
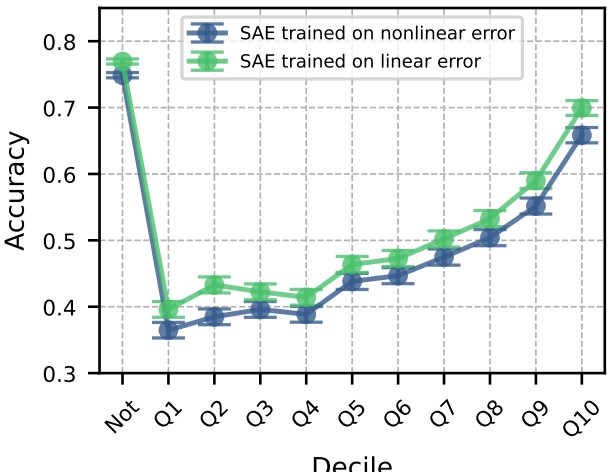

Figure 6: Violin plot of norm prediction tests for all SAEs (layer 12 Gemma 2B SAEs, layer 20 Gemma 9B SAEs, all Gemma Scope SAEs with $L_0$ closest to 60 across layers, and all Llama Scope SAEs). We plot the $R^2$ of a linear regression from $\mathbf{x}$ to to each random vector's norm squared.

Figure 7: Auto-interpretability results on SAEs trained on the linear and nonlinear components of $\texttt{SaeError}(\mathbf{x})$ on a width 16k, $L_0 \approx 60$, layer 20 Gemma Scope SAE. "Not" represents contexts that the SAE latent did not activate on, while each $Qi$ represents activating examples from decile $i$.

We run the following experiments in this section to understand how the linear and nonlinear components of SAE error differ:

1. In Section 5.1, we run the norm prediction test from Eq. (10) on different components of the error. We find that the norm of the nonlinear component is less predictable from activations than other components, implying that it might consist of fewer not-yet-learned features.

2. We train new SAEs directly on the linear and nonlinear components of error. The SAE trained on nonlinear error converges to higher reconstruction loss and produces less interpretable features.

3. We examine how much each component contributes to downstream model performance by intervening in the forward pass, and find that both components contribute proportionally to their size to increased cross entropy loss.

Overall, these experiments suggest that our split of SAE dark matter into these two categories is indeed a meaningful one.

## 5.1 Applying the Norm Prediction Test:

For our first experiment, we run the norm prediction test from Eq. (10) on five different random vectors: $\mathbf{x}$, $\texttt{Sae}(\mathbf{x})$, $\texttt{SaeError}(\mathbf{x})$, $\texttt{LinearError}(\mathbf{x})$, and $\texttt{NonlinearError}(\mathbf{x})$. The results are shown as a violin plot for each component across all 329 SAEs we experiment with in Section 4 (layer 12 Gemma 2B SAEs, layer 20 Gemma 9B SAEs, all Gemma Scope SAEs with $L_0$ closest to 60 across layers, and all Llama Scope SAEs) in Fig. 6 (the $\texttt{Sae}(\mathbf{x})$ bar can be seen as a summary of Fig. 2). There are one or two SAE with an outlier $R^2$ equal to or lower than 0; we omit these from the plot because they are likely due to numeric instability in the linear regression routine.

The intuition behind this test is that if a random vector consists of a sum of almost-orthogonal linear features from $\mathbf{x}$, then its norm should be linearly predictable from $\mathbf{x}$ (see Appendix B for more on this intuition).

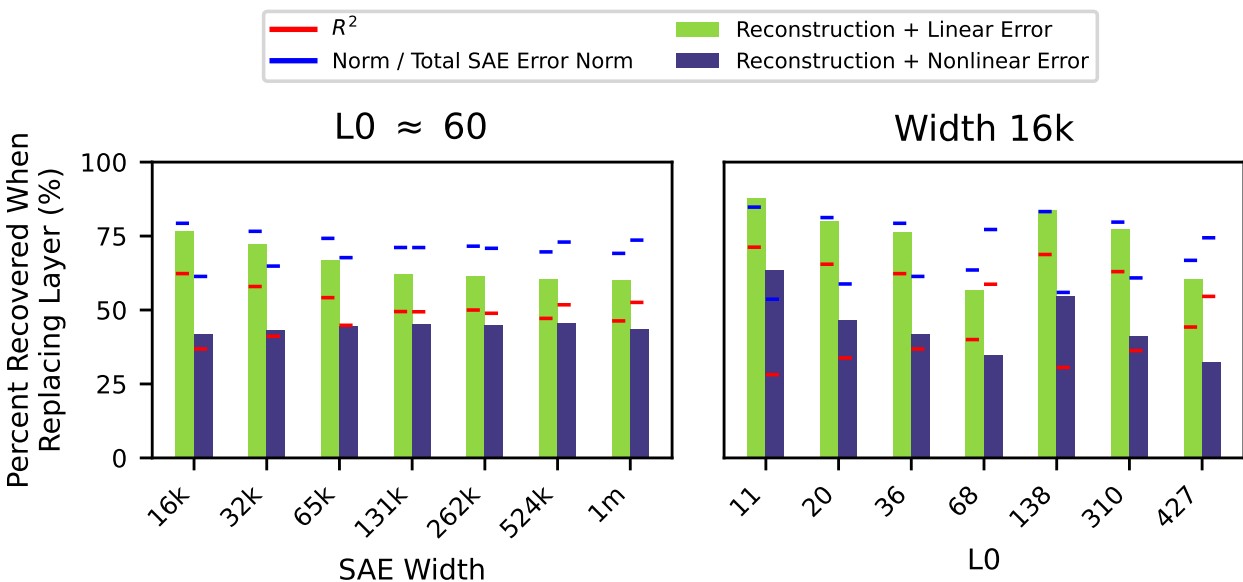

Figure 8: Results of intervening in the forward pass and replacing $\mathbf{x}$ with $\mathtt{Sae}(\mathbf{x}) + \mathtt{NonlinearError}(\mathbf{x})$ and $\mathtt{Sae}(\mathbf{x}) + \mathtt{LinearError}(\mathbf{x})$ during the forward pass on all layer 20 Gemma 9B SAEs with $L_0 \approx 60$ or width of 16k. Reported in percent of cross entropy loss recovered with respect to the difference between the same intervention with $\mathtt{Sae}(\mathbf{x})$ and with the normal model forward pass.

Firstly, we note that $\|\mathbf{x}\|_2^2$ can almost be perfectly predicted from $\mathbf{x}$. This is reassuring news for the linear representation hypothesis, as it implies that $\mathbf{x}$ may be well modeled as the sum of many one-dimensional features, at least from the perspective of this test. We also find that $\mathtt{NonlinearError}(\mathbf{x})$ (and $\mathtt{SaeError}(\mathbf{x})$, which consists partly of $\mathtt{NonlinearError}(\mathbf{x})$) has a notably lower score on this test than $\mathtt{LinearError}(\mathbf{x})$. This result suggests that $\mathtt{NonlinearError}(\mathbf{x})$ does not consist of a sparse sum of linear features from $\mathbf{x}$.

### 5.2 Training SAEs on $\mathtt{SaeError}(\mathbf{x})$ Components:

Another empirical test we run is training an SAE on $\mathtt{NonlinearError}(\mathbf{x})$ and $\mathtt{LinearError}(\mathbf{x})$. We choose a fixed Gemma 9B Gemma Scope layer 20 SAE with $16k$ latents and $L_0 \approx 60$ to generate $\mathtt{SaeError}(\mathbf{x})$ from. This SAE has nonlinear and linear components of the error approximately equal in norm and $R^2$ of the total $\mathtt{SaeError}(\mathbf{x})$ they explain, so it presents a fair comparison. We train SAEs to convergence (about 100M tokens) on each of these components of error and find that the SAE trained on $\mathtt{NonlinearError}(\mathbf{x})$ converges to a fraction of variance unexplained an absolute 5 percent higher than the SAE trained on the linear component of SAE error ($\approx 0.59$ and $\approx 0.54$ respectively).

We additionally examine the *interpretability* of the learned SAE latents using automated interpretability (this technique was first proposed by Bills et al. (2023) for interpreting neurons, and first applied to SAEs by Cunningham et al. (2023)). Specifically, we use the implementation introduced by Juang et al. (2024), where a language model (we use Llama 3.1 70b (AI@Meta, 2024)) is given top activating examples to generate an explanation, and then must use only that explanation to predict if the feature fires on a test context. Our results in Fig. 7 show that the SAE trained on linear error produces latents that are about an absolute 5% more interpretable across all activation firing deciles (we average results across 1000 random features for both SAEs, where for each feature use 7 examples in each of the 10 feature activation deciles as well as 50 negative examples, and show 95% confidence intervals).

### 5.3 Downstream Cross Entropy Loss of $\mathtt{SaeError}(\mathbf{x})$ Components:

A common metric used to test SAEs is the percent of cross entropy loss recovered when the SAE reconstruction is inserted into the model in place of the original activation versus an ablation baseline (see e.g. Bloom

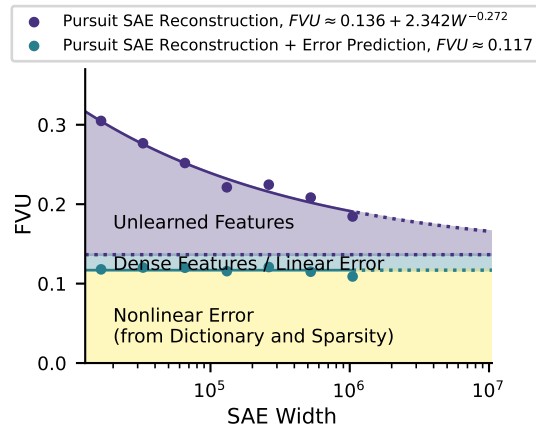
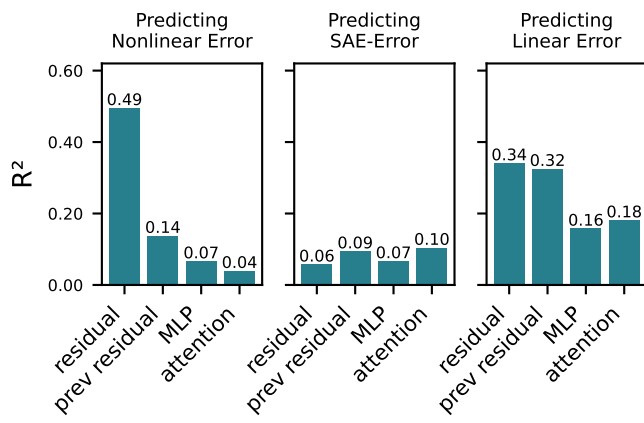

(a) SAE error breakdown vs. SAE width for inference time optimized reconstructions from Gemma Scope $L_0 \approx 60$ dictionaries.

(b) The $R^2$ of linearly predicting parts of SAE error from the SAE reconstructions of adjacent model components, layer 20 Gemma Scope $L_0 \approx 60$, $16k$ SAE width.

Figure 9: Investigations towards reducing nonlinear SAE error.

(2024)). We modify this test to specifically examine the different components of `SaeError(x)`: we compare the percent of the cross entropy loss recovered when replacing $\mathbf{x}$ with `Sae(x)` plus either `LinearError(x)` or `NonlinearError(x)` to the baseline of inserting just `Sae(x)` in place of $\mathbf{x}$ on all layer 20 Gemma 9B SAEs with $L_0 \approx 60$ or width of 16k. To estimate how much each component "should" recover, we use two metrics: the average norm of the component relative to the total norm of `Sae(x)` and the percent of the variance that the component recovers between `Sae(x)` and $\mathbf{x}$. The results, shown in Fig. 8, show that for the most part these metrics are reasonable predictions for both types of error. That is, both `NonlinearError(x)` and `LinearError(x)` proportionally contribute to the SAE's increase in downstream cross entropy loss, with possibly a slightly higher contribution than expected for `LinearError(x)`.

## 6    Reducing `NonlinearError(x)`

In past sections we identify `NonlinearError(x)` and `LinearError(x)` and argue that `NonlinearError(x)` likely does not consist of not-yet-learned linear features. This may be a problem for scaling SAEs farther: if a part of SAE error persistently consists of nonlinear transformations of linear features, these may never be able to be learned by SAEs. Thus, in this section, we investigate two approaches for reducing `NonlinearError(x)`:

1. Using a more powerful encoder at inference time. This approach has limited success, reducing total FVU by 3-5% but leaving `FVU_nonlinear` almost unchanged

2. Attempting to predict `NonlinearError(x)` from SAE reconstructions of adjacent model components - this approach is more successful; on the SAE we test on, `Sae(x)` predicts 50% of the nonlinear error, and previous model components predict between 4% and 14% of nonlinear error, although only between 6% and 10% of the overall SAE error.

### 6.1    Using a More Powerful Encoder

Our first approach for reducing nonlinear error is to try improving the encoder. We use a recent approach suggested by Smith (2024a): applying a greedy inference time optimization (ITO) algorithm called gradient pursuit to a frozen learned SAE decoder matrix. We implement and run ITO on all layer 20 Gemma Scope $9b$ SAEs closest to $L_0 \approx 60$. For each example $\mathbf{x}$ with reconstruction `Sae(x)`, we use the gradient pursuit implementation with an $L_0$ exactly equal to the $L_0$ of $\mathbf{x}$ in the original `Sae(x)`.

Using these new reconstructions of $\mathbf{x}$, we repeat Eq. (11) and do a linear transformation from $\mathbf{x}$ to the inference time optimized reconstructions. We then regenerate the same scaling plot as Fig. 1 and show this

figure in Fig. 9a. Our first finding is that pursuit indeed decreases the total FVU of $\texttt{Sae}(\mathbf{x})$ by 3 to 5%; as Smith (2024a) only showed an improvement on a small 1 layer model, to the best of our knowledge we are the first to show this result on state of the art SAEs. Our most interesting finding, however, is that the $\texttt{FVU}_{\texttt{nonlinear}}$ stays almost constant when compared to the original SAE scaling in Fig. 1. We hypothesize that this might happen because ITO reduces "easy" linear errors like feature shrinkage. In Fig. 1, we plot the additional reduction in $\texttt{FVU}_{\texttt{nonlinear}}$ as the contribution of encoder error; because $\texttt{FVU}_{\texttt{nonlinear}}$ stays almost constant, this section is very narrow.

## 6.2 Predicting SAE Errors Between SAEs:

For the Gemma 2 architecture at the locations the SAEs are trained on, each residual activation can be decomposed in terms of prior components:

$$\texttt{Resid}_{\texttt{layer}} = \texttt{MlpOut}_{\texttt{layer}} + \texttt{RMSNorm}(\texttt{O}_{\texttt{proj}}(\texttt{AttnOut}_{\texttt{layer}})) + \texttt{Resid}_{\texttt{layer}-1} \qquad (16)$$

We focus on $layer = 20$ and Gemma 2 9B, and thus use the layer 19 attention and MLP Gemma Scope SAEs. For all components we use the width $16k$ SAEs with $L_0 \approx 60$. In Fig. 9b, we plot the $R^2$ of a regression from the SAE output corresponding to each of these right hand side components to each of the different components of an SAE trained on $\texttt{Resid}_{\texttt{layer}}$ ($\texttt{SaeError}(\mathbf{x})$, $\texttt{LinearError}(\mathbf{x})$, and $\texttt{NonlinearError}(\mathbf{x})$).

We find that we can explain a small amount (up to $\approx 10\%$) of total $\texttt{SaeError}(\mathbf{x})$ using previous components, which may be immediately useful for circuit analysis. We also can explain large parts of $\texttt{NonlinearError}(\mathbf{x})$ with prior components as well. While $\texttt{Sae}(\mathbf{x})$ explains 50% of the variance in the nonlinear error, this may be somewhat misleading, as the nonlinear error is partially a function of $\texttt{Sae}(\mathbf{x})$:

$$\texttt{NonlinearError}(\mathbf{x}) = \texttt{SaeError}(\mathbf{x}) - \texttt{LinearError}(\mathbf{x})$$
$$= (\mathbf{x} - \texttt{Sae}(\mathbf{x})) - \texttt{LinearError}(\mathbf{x})$$

These results mean that we might be able to explain some of the SAE Error using a circuits level view, but that overall there are still large parts of each error component unexplained.

## 7 Conclusion

The fact that SAE error can be predicted and analyzed at all is surprising. Thus, our findings are intriguing evidence that SAE errors, and not just SAE reconstructions, are worthy of analysis, and we hope that it inspires further work in decomposing SAE error.

Concretely, we believe that our study of error has already discovered a number of promising directions for future SAE research. The discovery that SAE errors can be significantly predicted from model activations hints that some subspaces may resist being learned and points toward potential architectural innovations such as incorporating low-rank dense side channels. The predictable relationship between small and large SAE errors may also streamline experimentation with novel SAE architectures by allowing researchers to efficiently forecast scaling behavior through small-scale trials. Additionally, our demonstration that prior SAE outputs can explain part of SAE error has immediate practical implications for circuit analysis, which currently relies on large noise terms that complicate circuit interpretation.

At a higher level, the presence of constant nonlinear error at a fixed sparsity as we scale implies that scaling SAEs may not be the only (or best) way to explain more of model behavior. Future work might explore alternative penalties besides sparsity or new ways to learn better dictionaries. Ultimately, we believe that there is still room to make SAEs better, not just bigger.

### Acknowledgments

Our work benefited greatly from the thoughtful comments and discussions provided by (in alphabetical order) Joseph Bloom, Lauren Greenspan, Jake Mendel, and Eric Michaud. We are deeply appreciative of their contributions. This work was supported by Erik Otto, Jaan Tallinn, the Rothberg Family Fund for

Cognitive Science, and IAIFI through NSF grant PHY-2019786. JE was supported by the NSF Graduate Research Fellowship (Grant No. 2141064). LS was supported by the Long Term Future Fund.

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

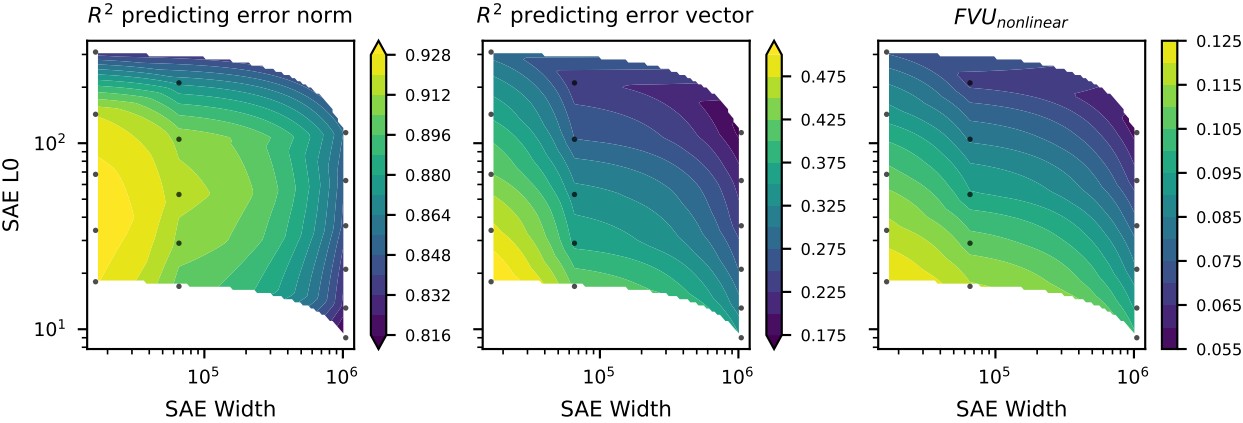

(a) Linear prediction results for layer 5 Gemma 2 2B SAEs from Gemma Scope.

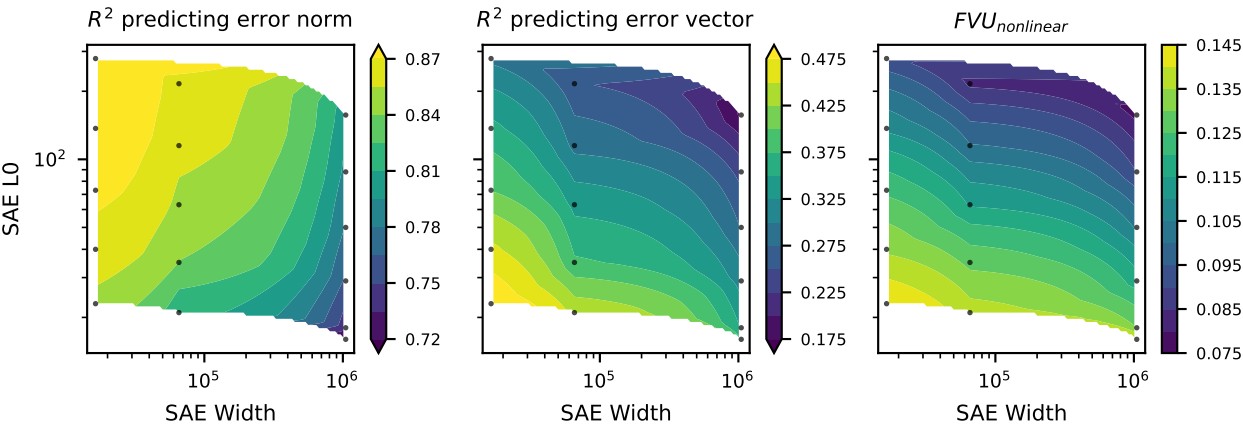

(b) Linear prediction results for layer 19 Gemma 2 9B SAEs from Gemma Scope.

Figure 10: Results of linearly predicting SAE error norm and SAE error from model activations on Gemma 2 2B layer 5 (**top**) and Gemma 2 2B layer 19 (**bottom**). The right plots show the $R^2$ of predicting SAE error norms (see Eq. (10)), the middle plots show the $R^2$ of predicting SAE error vectors (see Eq. (11), and the right plots show $1 - R^2$ of predicting model activations given the SAE reconstruction and the SAE error vector prediction. Unlike for the middle layers shown in the main body, $\texttt{FVU}_{\texttt{nonlinear}}$ decreases some as width is scaled, although the sparsity of the SAEs in the space makes this harder to verify.

# A  Extra Error Prediction Experiments

## A.1  Note on Feature Shrinkage

Earlier SAE variants were prone to *feature shrinkage*: the observation that $\texttt{Sae}(\mathbf{x})$ systematically undershot $\mathbf{x}$. Current state of the art SAE variants (e.g. JumpReLU SAEs and TopK SAEs, which we examine in this work), are less vulnerable to this problem, although we still find that Gemma Scope reconstructions have about a 10% smaller norm than $\mathbf{x}$. One potential concern is that the $\mathbf{b}^*$ in Eq. (11) that we learn is merely predicting this shrinkage. If this was the case, then the cosine similarity of the linear error prediction $(\boldsymbol{b}^*)^T \cdot \mathbf{x}$ with $\mathbf{x}$ would be close to 1; however, in practice we find that it is around 0.5, so $\boldsymbol{b}^*$ is indeed doing more than predicting shrinkage.

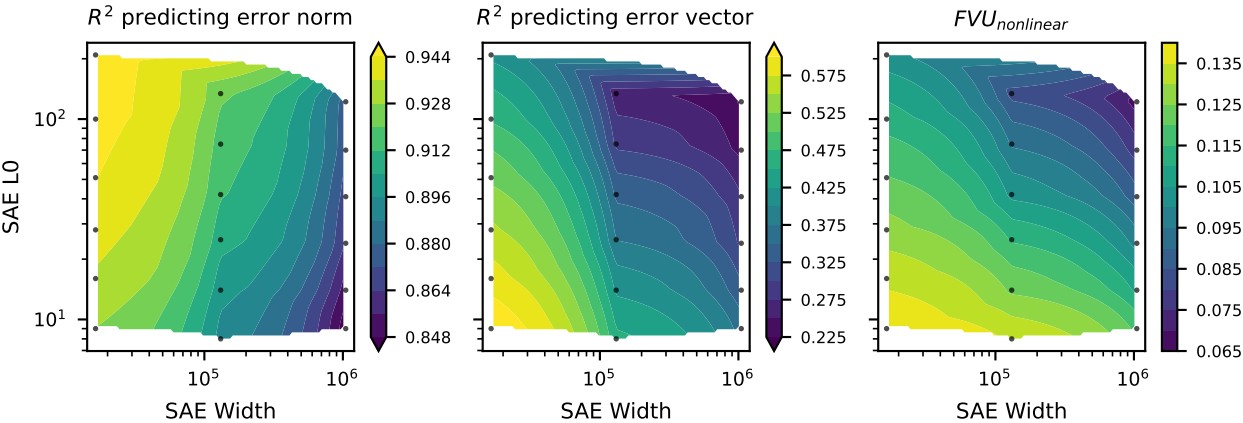

(a) Linear prediction results for layer 9 Gemma 2 9B SAEs from Gemma Scope.

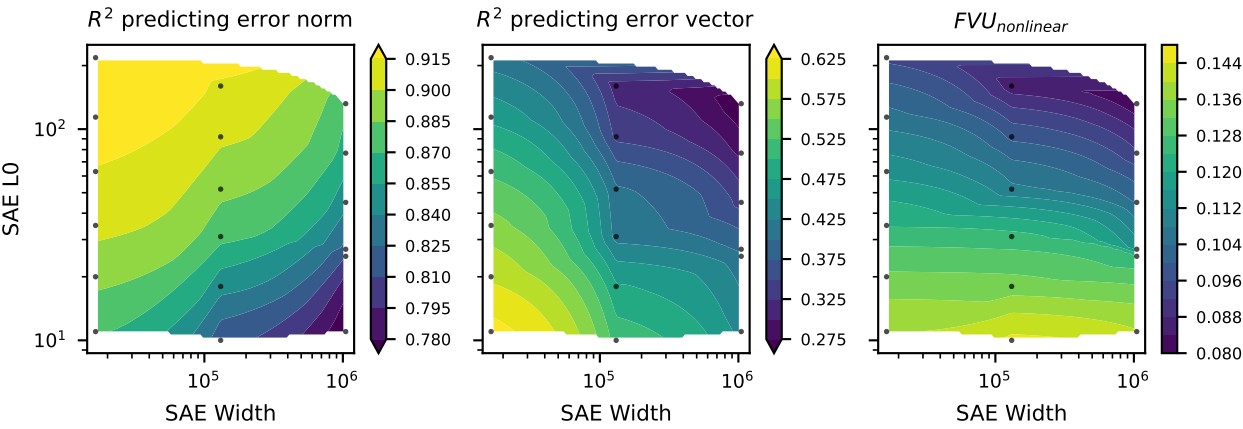

(b) Linear prediction results for layer 31 Gemma 2 9B SAEs from Gemma Scope.

Figure 11: Results of linearly predicting SAE error norm and SAE error from model activations on Gemma 2 9B layer 9 (**top**) and Gemma 2 9B layer 31 (**bottom**). The right plots show the $R^2$ of predicting SAE error norms (see Eq. (10), the middle plots show the $R^2$ of predicting SAE error vectors (see Eq. (11), and the right plots show $1 - R^2$ of predicting model activations given the SAE reconstruction and the SAE error vector prediction. Unlike for the middle layers shown in the main body, $\text{FVU}_{\text{nonlinear}}$ decreases some as width is scaled, although the sparsity of the SAEs in the space makes this harder to verify.

## A.2 Additional Error Prediction Plots

In Section 7 and Section 7, we show the accuracy of SAE error norm predictions, SAE error vector predictions, and $\text{FVU}_{\text{nonlinear}}$ for layers 5 and 19 of Gemma 2 2B and layers 9 and 31 of Gemma 2 9B. As described in the main text, we find broadly similar results for SAE error norm and error vector prediction at these layers, but find that $\text{FVU}_{\text{nonlinear}}$ decreases some as we increase SAE width at a fixed $L_0$, although this result is uncertain because of the sparsity of Gemma Scope SAEs at these layers.

## A.3 Norm Prediction Baselines

In Fig. 13, we run a linear regression from different model variables to $\|\texttt{SaeError}(\mathbf{x})\|$ across layers on Gemma Scope 9B for SAEs of size 131k. We find that is is not "easy" to predict SAE error norm, especially at later layers; the token identity, SAE L0, activation norm, and model loss all do significantly worse than using the full activation. It is interesting to note that at the first few layers, token identity does better at

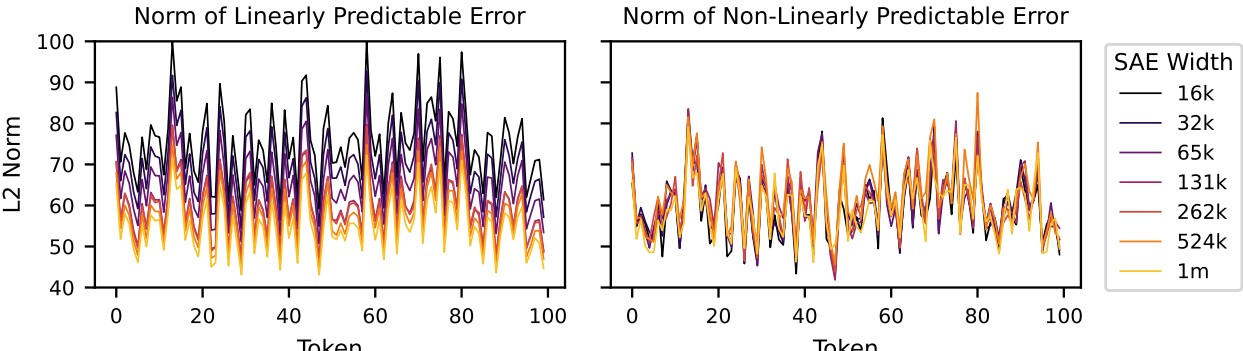

Figure 12: Per-token breakdown of linearly predictable and non-linearly predictable SAE error across SAE scale. We show the same tokens as in Fig. 5. The norm of linear error decreases with SAE width, whereas the norm of nonlinear error stays mostly constant.

predicting SAE error than a probe of the activations; this is perhaps not surprising, since recent results from e.g. Lad et al. (2024) show that very early layers primarily operate on a per token level.

### A.4 Breaking Apart Error Per Token

In Fig. 12, we show the same subset of tokens as in Fig. 5, but now broken apart into linearly predictable and non-linearly predictable components. That is, we learn $\boldsymbol{b}^*$ for each SAE as in Eq. (11), and then plot the norm of $\boldsymbol{b}^* \cdot \mathbf{x}$ at the norm of the linearly predictable error on the left, and plot the norm of $\texttt{SaeError}(\mathbf{x}) - \boldsymbol{b}^* \cdot \mathbf{x}$ as the norm of the non-linearly predictable error on the right. We see that the linearly predictable error decreases as we scale SAE width, but the non-linearly predictable error mostly stays constant. This is especially interesting because the result in Fig. 1 just found this on an average level, whereas here we find the same result holds on a per-token level.

## B  Modeling Activations

In this section, we seek to explain why we see a difference between linear and nonlinear error in the main body of the paper. We will adopt the *weak* linear hypothesis (Smith, 2024b), a generalization of the linear representation hypothesis which holds only that *some* features in language models are represented linearly. Thus we have

$$\mathbf{x} = \sum_{i=0}^{n} \mathrm{w}_i \boldsymbol{y}_i + \texttt{Dense}(\mathbf{x}) \tag{17}$$

for linear features $\{\boldsymbol{y}_1, \ldots, \boldsymbol{y}_n\}$ and random vector $\mathbf{w} \in \mathbb{R}^n$, where $\mathbf{w}$ is sparse ($\|\mathbf{w}\|_1 \ll d$) and $\texttt{Dense}(\mathbf{x})$ is a random vector representing the dense component of $\mathbf{x}$. $\texttt{Dense}(\mathbf{x})$ might be Gaussian noise, nonlinear features as described by Csordás et al. (2024), or anything else not represented in a low-dimensional linear subspace.

Before we proceed with our analysis, we note that there is a rich sparse coding literature studying dictionary learning in the setting with mixed sparse and dense signals. For example, in a classic work Candès et al. (2011) propose Robust Principal Component Analysis for decomposing data matrices into sparse and dense components, although this is not directly applicable to our setting of trying to learn an autoencoder. More recently, Tasissa et al. propose a dictionary learning technique for data exactly modeled as in Eq. (17). In our work, we focus on studying existing SAEs and hypothesizing why they fail, so these works are not immediately applicable, but we are excited to see future work that applies these combined sparse-dense autoencoders to language model activations..

Say our SAE has $m$ latents. Since by assumption $\texttt{Dense}(\mathbf{x})$ cannot be represented in a low-dimensional linear subspace, the sparsity limited SAE will not be able to learn it. Thus, we will assume that the SAE learns

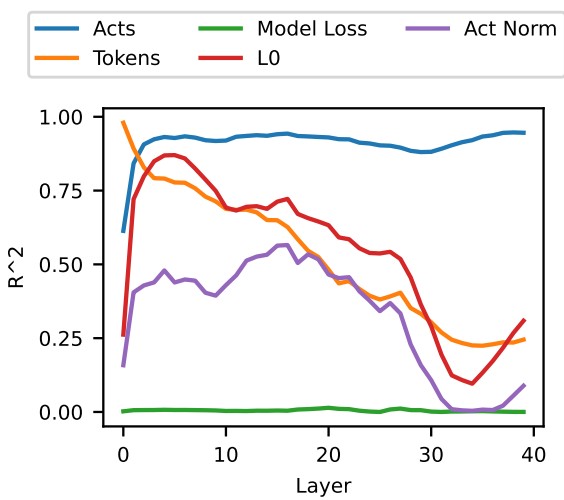

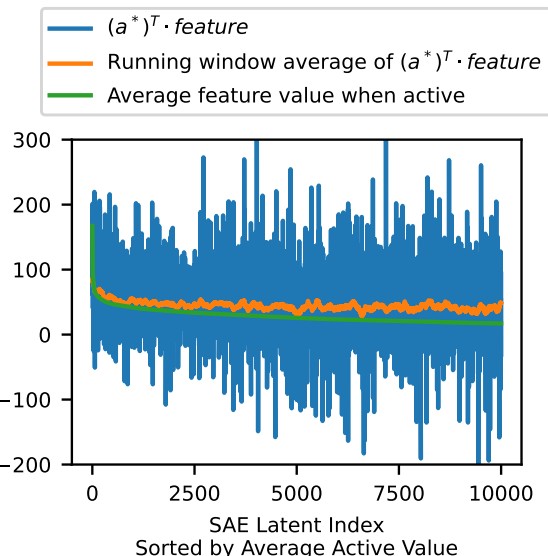

Figure 13: $R^2$ for linear regressions of SAE error norms with different regressors. We run on Gemma Scope 9B SAEs of size $131k$ with $L_0 \approx 60$. Activations perform the best except on the first few layers.

Figure 14: Average SAE latent activation and dot product of the latent with the learned norm prediction vector $\boldsymbol{a}^*$ for the Gemma Scope layer 20, width $131k$, $L_0 = 62$ SAE. We also plot a smoothed version of this dot product with a smoothed window of 10.

only the $m$ most common features $\boldsymbol{y}_0, \ldots, \boldsymbol{y}_{m-1}$. We will also assume that the SAE introduces some error when making this approximation, and instead learns $\hat{\boldsymbol{y}}_i$ and $\hat{w}_i$. Thus we have

$$\texttt{Sae}(\mathbf{x}) = \sum_{i=0}^{m} \hat{w}_i \hat{\boldsymbol{y}}_i \tag{18}$$

$$\texttt{SaeError}(\mathbf{x}) = \texttt{Dense}(\mathbf{x}) + \left( \sum_{i=0}^{m} \hat{w}_i \hat{\boldsymbol{y}}_i - \sum_{i=0}^{m} w_i \boldsymbol{y}_i \right) + \sum_{i=m}^{n} w_i \boldsymbol{y}_i \tag{19}$$

We finally define $\texttt{Introduced}(\mathbf{x}) := \sum_{i=0}^{m} \hat{w}_i \hat{\boldsymbol{y}}_i - \sum_{i=0}^{m} w_i \boldsymbol{y}_i$, so we have

$$\texttt{SaeError}(\mathbf{x}) = \texttt{Dense}(\mathbf{x}) + \texttt{Introduced}(\mathbf{x}) + \sum_{i=m}^{n} w_i \boldsymbol{y}_i \tag{20}$$

### B.1 Analyzing Error Norm Prediction

We will first analyze Eq. (10), the learned probe from $\mathbf{x}$ to $\|\texttt{SaeError}(\mathbf{x})\|_2^2$. First, we claim that given a vector $\mathbf{x}$, if $\mathbf{x}$ is a sparse sum of orthogonal vectors, then by basic linear algebra there exists a perfect prediction vector $\boldsymbol{a}$ such that $\boldsymbol{a}^T \mathbf{x} \approx |\mathbf{x}|_2^2$ (in other words, the norm squared of $\mathbf{x}$ can be linearly predicted from $\mathbf{x}$). The proof of this claim is in Appendix C.1; the intuition is that we can set the probe vector $\boldsymbol{a}^*$ to the sum of the vectors $\boldsymbol{y}_i$ weighted by their average weight $E(w_i)$.

When $\mathbf{x}$ is instead a sparse sum of *non-orthogonal* vectors, as it partly is in Eq. (17) and Eq. (20), this proof is no longer true, but we now argue that a similar intuition holds. If the $\boldsymbol{y}_i$ are almost orthogonal (formally *coherence* $< \epsilon$, see (Foucart & Rauhut, 2013)) and do not activate much at the same time, then a probe vector again equal to the sum of vectors $\boldsymbol{y}_i$ weighted by their average value $E(w_i)$ will be a good approximate prediction. Indeed, when we try predicting $\|\texttt{Sae}(\mathbf{x})\|_2^2$ from $\texttt{Sae}(\mathbf{x})$ (which is a sparse sum of

known almost orthogonal vectors of a similar distribution to the true SAE vectors), we find that indeed the linear probe that is learned is approximately equal to this sum (see Appendix C.2).

Thus, we can now neatly explain why we can predict the norms of SAE errors: they mostly consist of almost orthogonal sparsely occuring not yet learned SAE features! We further can explain why larger SAEs have less predictable error norms: since $m$ is larger, there is a larger component in the error of not-as-linearly-predictable $\mathtt{Dense}(\mathbf{x})$ and $\mathtt{Introduced}(\mathbf{x})$.

## B.2 Analyzing Error Vector Prediction

We will now analyze Eq. (11), the learned transformation from $\mathbf{x}$ to $\mathtt{SaeError}(\mathbf{x})$, with our model of SAE error from Eq. (20). We assume that $\mathtt{Introduced}(\mathbf{x})$ cannot be approximated at all as a linear function of $\mathbf{x}$. This is a reasonable assumption since the SAE is a nonlinear function of $\mathbf{x}$, but if indeed some of $\mathtt{Introduced}(\mathbf{x})$ can be approximated in this way, then we will underestimate the amount of $\mathtt{Introduced}(\mathbf{x})$ and therefore overestimate the amount of $\mathtt{Dense}(\mathbf{x})$.

If $\mathtt{Dense}(\mathbf{x}) + \sum_{i=m}^{n} \mathtt{w}_i \boldsymbol{y}_i$ is contained in a linear subspace of $\mathbf{x}$ orthogonal to $\sum_{i=0}^{m} \mathtt{w}_i \boldsymbol{y}_i$, then $\mathtt{Dense}(\mathbf{x})$ is part of the linearly explainable error, and the error of the transformation $\boldsymbol{b}^*$ exactly equals $\mathtt{Introduced}(\mathbf{x})$ (since the transformation is just exactly this orthogonal linear subspace). However, if such an orthogonal linear subspace does not exist, the optimal linear transform will only be able to reconstruct part of $\mathtt{Dense}(\mathbf{x}) + \sum_{i=m}^{n} \mathtt{w}_i \boldsymbol{y}_i$, and the percent of variance left unexplained by the regression will be an upper bound on the true variance explained by $\mathtt{Introduced}(\mathbf{x})$. We also note that if this linear transform indeed recovers $\mathtt{Dense}(\mathbf{x})$ and absent features but fails to recover $\mathtt{Introduced}(\mathbf{x})$, we can use it to estimate $\mathtt{Dense}(\mathbf{x})$: the difference between the variance explained by $\mathtt{Sae}(\mathbf{x})$ and the variance explained by $\mathbf{x} - (\mathtt{Sae}(\mathbf{x}) + \mathbf{b}^* \cdot \mathbf{x})$ will approach $\mathtt{Dense}(\mathbf{x})$ as $m \to \infty$.

Thus, our ability to estimate $\mathtt{Introduced}(\mathbf{x})$ and $\mathtt{Dense}(\mathbf{x})$ using $\boldsymbol{b}^*$ depends on how well a linear transform works to predict $\mathtt{Dense}(\mathbf{x})$ and $\sum_{i=m}^{n} \mathtt{w}_i \boldsymbol{y}_i$. Although we do not have access to the ground truth vectors $\boldsymbol{y}_i$, we *can* replace $\mathbf{x}'$ with a similar distribution of vectors that we *do* have access to, using the same trick as above. Given an SAE, we replace $\mathbf{x}$ with $\mathbf{x}' = \mathtt{Sae}(\mathbf{x})$. $\mathbf{x}'$ has the useful property that it is a sparse linear sum of vectors (the ones that the SAE learned), and the distribution of these vectors and their weights are similar to that of the true features $\boldsymbol{y}_i$. We now pass $\mathbf{x}'$ back through the SAE and can control all of the quantities we are interested in: we can vary $m$ by masking SAE dictionary elements, simulate $\mathtt{Dense}(\mathbf{x}')$ by adding Gaussian noise to $\mathbf{x}'$, and simulate $\mathtt{Introduced}(\mathbf{x})$ by adding Gaussian noise to $\mathtt{Sae}(\mathbf{x}')$.

We run this synthetic setup with a Gemma Scope layer 20 SAE (width $16k$, $L_0 \approx 68$) in Appendix C.3, and find that indeed, estimated $\mathtt{Dense}(\mathbf{x}')$ is highly correlated with the amount of Gaussian noise added to $\mathbf{x}'$ and $\mathtt{Introduced}(\mathbf{x}')$ is highly correlated with the amount of Gaussian noise added to $\mathtt{Sae}(\mathbf{x}')$ (see Ta-

Table 1: Correlation matrix between synthetic noise and estimated errors.

|  | Estimated $\mathtt{Dense}(\mathbf{x}')$ | Estimated $\mathtt{Introduced}(\mathbf{x}')$ |
|---|---|---|
| $\mathbf{x}'$ Noise | 0.9842 | 0.1417 |
| $\mathtt{Sae}(\mathbf{x}')$ Noise | 0.0988 | 0.9036 |

ble 1). However, note that because $\mathbf{x}'$ noise is also slightly correlated with estimated $\mathtt{Introduced}(\mathbf{x}')$, it is possible that some of the contribution to the estimated nonlinear error is from $\mathtt{Dense}(\mathbf{x}')$.

Thus, we again now have a potential explanation for our initial results: we can predict error vectors because they consist in large part of not yet learned linear features in an almost orthogonal subspace of $\mathbf{x}$, we can predict a smaller portion of larger SAE errors because the number of these linear features go down with SAE width, and the horizontal line in Fig. 1 is because $\mathtt{Dense}(\mathbf{x})$ and $\mathtt{Introduced}(\mathbf{x})$ are mostly constant. Furthermore, we can hypothesize from the correlations om Table 1 that the linearly predictable component of SAE error consists mostly of not yet learned features and $\mathtt{Dense}(\mathbf{x})$, while the component that is not linearly predictable consists mostly of $\mathtt{Introduced}(\mathbf{x})$. We will explore this hypothesis in Section 5.

### B.3 Analyzing Per-Token Scaling Predictions

Finally, we provide a simple explanation for why per-token SAE errors are highly predictable between SAEs of different sizes. For this, we only need Eq. (20). Since $\texttt{Dense}(\mathbf{x})$ and $\texttt{Introduced}(\mathbf{x})$ stay mostly constant as $m$ increases, for large $m$ the SAE error stays mostly constant because it is primarily determined by these components. Thus, since $m = 16k$ is already large, a linear prediction that is just a slightly smaller version of the current error performs well. Additionally, this reasoning suggests a natural experiment: if we can predict $\texttt{Introduced}(\mathbf{x})$ on a per-token level (which we hypothesize we can do with the non-linearly predictable component of SAE error), we may be able to better predict the floor of SAE scaling and therefore better predict larger SAE errors; we run this experiment in Appendix E, where we find an affirmative answer.

## C  More Info on Modeling Activations

### C.1  Proof of Claim from Appendix B.1

Say we have a set of $m$ unit vectors $\boldsymbol{y}_1, \boldsymbol{y}_2, \ldots, \boldsymbol{y}_m \in \mathbb{R}^d$. We will call these "feature vectors". Define $\boldsymbol{Y} \in \mathbb{R}^{d \times m}$ as the matrix with the feature vectors as columns. We then define the Gram matrix $\mathbf{G_Y} \in \mathbb{R}^{m \times m}$ of dot products on $\mathbf{Y}$:

$$(\mathbf{G_Y})_{ij} = (\mathbf{Y}^T \mathbf{Y})_{ij} = \boldsymbol{y}_i \cdot \boldsymbol{y}_j$$

We now will define a random column vector $\mathbf{x}$ that is a weighted positive sum of the $m$ feature vectors, that is, $\mathbf{x} = \sum_i w_i \boldsymbol{y}_i$ for a non-negative random vector $\mathbf{w} \in \mathbb{R}^m$. We say feature vector $\boldsymbol{y}_i$ is active if $w_i > 0$. We now define the autocorrelation matrix $\mathbf{R_w} \in \mathbb{R}^{m \times m}$ for $\mathbf{w}$ as

$$\mathbf{R} = \mathbb{E}(\mathbf{w}\mathbf{w}^T).$$

We are interested in breaking down $\mathbf{x}$ into its components, so we define a random matrix $\mathbf{X}$ as $\mathbf{X}_{ij} = w_j \mathbf{Y}_{ij}$, i.e. the columns of $\boldsymbol{Y}$ multiplied by $\mathbf{w}$. We can now define the Gram matrix $\mathbf{G_X} \in \mathbb{R}^{m \times m}$:

$$(\mathbf{G_X})_{ij} = (\mathbf{X}^T \mathbf{X})_{ij} = w_i w_j \boldsymbol{y}_i \cdot \boldsymbol{y}_j$$
$$\mathbf{G_X} = (\mathbf{w}\mathbf{w}^T) \odot \mathbf{G_Y}$$
$$\mathbb{E}(\mathbf{G_X}) = \mathbf{R_w} \odot \mathbf{G_Y},$$

where $\odot$ denotes Schur (elementwise) multiplication. The intuition here is that the expected dot product between columns of $\mathbf{X}$ depends on the dot product between the corresponding columns of $\mathbf{Y}$ and the correlation of the corresponding elements of the random vector.

We will now examine the L2 norm of $\mathbf{x}$:

$$\|\mathbf{x}\|_2^2 = \sum_{ij} w_i w_j \mathbf{y}_i \mathbf{y}_j$$
$$= \left\|(\mathbf{w}^T \mathbf{w}) \odot \mathbf{G_Y}\right\|_F^2 = \mathrm{Tr}(\mathbf{w}\mathbf{w}^t \mathbf{G_Y}) = \mathbf{w}\mathbf{G_Y}\mathbf{w}^T$$

We can also take the expected value:

$$\mathbb{E}(\|\mathbf{x}\|_2^2) = \mathrm{Tr}(\mathbf{R_w}\mathbf{G_Y})$$

Our goal is to find a direction $\boldsymbol{a} \in \mathbb{R}^d$ that when dotted with $\mathbf{x}$ predicts $\|\mathbf{x}\|_2^2$. In other words, we want to find $\mathbf{a}$ such that

$$\|\mathbf{x}\|_2^2 \approx \boldsymbol{a}^T \mathbf{x} = \boldsymbol{a}^T \sum_i w_i \boldsymbol{y}_i = \boldsymbol{a}^T \mathbf{Y}\mathbf{w}$$

Combining equations, we want to find $\mathbf{a}$ such that

$$\boldsymbol{a}^T \mathbf{Y}\mathbf{w} \approx \|\mathbf{x}\|_2^2 = (vw^T \mathbf{G_Y}\mathbf{w})$$

Let us first consider the simple case where for all $i \neq j$, $y_i$ and $y_j$ are perpendicular. Then our goal is to find $\boldsymbol{a}$ such that

$$\boldsymbol{a}^T \mathbf{Y}\mathbf{w} \approx \mathrm{Tr}(\mathbf{w}\mathbf{G_Y}\mathbf{w}^T) = \sum_i \langle y_i, y_i \rangle \, \mathrm{w}_i^2 = \sum_i \mathrm{w}_i^2 = \|\mathbf{w}\|_2^2 = \mathbf{w}^T\mathbf{w}$$

Since all of the $y_i$ are perpendicular, WLOG we can write $\boldsymbol{a} = \sum_i b_i \boldsymbol{y}_i + \boldsymbol{c}$ for a vector $\boldsymbol{c} \in \mathbb{R}^d$ perpendicular to all $\boldsymbol{y}_i$ and a vector $\boldsymbol{b} \in \mathbb{R}^m$. Then we have

$$\boldsymbol{a}^T \mathbf{Y}\mathbf{w} = \left( \sum_i b_i \boldsymbol{y}_i + \boldsymbol{c} \right)^T \mathbf{Y}\mathbf{w}$$
$$= \boldsymbol{b}^T\mathbf{w}$$

Since ordinary least squares produces an unbiased estimator, we know that if we use ordinary least squares to solve for $\boldsymbol{b}$, $\mathbb{E}(\boldsymbol{b}^T\mathbf{w}) = \mathbb{E}(\mathbf{w}^T\mathbf{w})$. Thus,

$$\sum_i b_i \mathbb{E}(w_i) = \sum_i \mathbb{E}(w_i^2)$$
$$b_i = \mathbb{E}(w_i^2)/\mathbb{E}(w_i)$$

Now that we have $b_i$, we can solve for the correlation coefficient between $\boldsymbol{a}^T\mathbf{x} = \boldsymbol{b}^T\mathbf{w}$ and $\|\mathbf{x}\|_2^2 = \mathbf{w}^T\mathbf{w}$. This gets messy when using general distributions, so we focus on a few simple cases.

The first is the case where each $\mathrm{w}_i$ is a scaled independent Bernoulli distribution, so $w_i$ is $s_i$ with probability $p_i$ and 0 otherwise. Then $b_i = s_i$. We also have that $\mathbb{E}(\mathbf{w}^T\mathbf{w}) = \mathbb{E}(\boldsymbol{b}^T\mathbf{w}) = \sum_i s_i^2 p_i = \mu$.

$$\rho = \frac{\mathbb{E}(\boldsymbol{b}^T\mathbf{w}\mathbf{w}^T\mathbf{w}) - \mu^2}{\sqrt{\mathbb{E}(\mathbf{w}^T\mathbf{w}\mathbf{w}^T\mathbf{w}) - \mu^2}\sqrt{\mathbb{E}(\mathbf{b}^T\mathbf{w}\boldsymbol{b}^T\mathbf{w}) - \mu^2}}$$
$$= \frac{\sum_i s_i^4(p_i - p_i^2)}{\sqrt{\sum_i s_i^4(p_i - p_i^2)}\sqrt{\sum_i s_i^4(p_i - p_i^2)}} = 1$$

That is, for Bernoulli variables, $\mathbf{x} = \sum_i s_i \boldsymbol{y}_i$ is a perfect regression vector.

The second is the case when each $\mathrm{w}_i$ is an independent Poisson distribution with parameter $\lambda_i$. Then $\mathbb{E}(\mathrm{w}_i) = \lambda_i$ and $\mathbb{E}(\mathrm{w}_i^2) = \lambda_i^2 + \lambda_i$, so $b_i = \lambda_i + 1$. We also have that $\mathbb{E}(\mathbf{w}^T\mathbf{w}) = \mathbb{E}(\boldsymbol{b}^T\mathbf{w}) = \sum_i \lambda_i^2 + \lambda_i = \mu$. Finally, we will use the fact that $\mathbb{E}(\mathrm{w}_i^3) = \lambda_i^3 + 3\lambda_i^2 + \lambda_i$ and $\mathbb{E}(\mathrm{w}_i^4) = \lambda^4 + 6\lambda^3 + 7\lambda^2 + \lambda$. Then via algebra we have that

$$\rho = \frac{\sum_i 2\lambda_i^3 + 3\lambda_i^2 + \lambda_i}{\sqrt{\sum_i 4\lambda^3 + 6\lambda_i^3 + \lambda_i}\sqrt{\sum_i \lambda_i^3 + 2\lambda_i^2 + \lambda_i}}$$

For the special case $\lambda_i = 1$, we then have

$$\rho = \frac{6}{\sqrt{66}} \approx 0.73$$

## C.2 Empirical Norm Prediction

In this experiment, we aim to determine to what extent our analysis in Appendix B.1 holds true in practice on almost orthogonal true SAE features. Thus, we use a random vector that we can control: $\mathtt{Sae}(\mathbf{x})$. Specifically, we learn a probe $\boldsymbol{a}^*$ for the Gemma Scope layer 20, width $131k$, $L_0 = 62$ SAE as in Eq. (10), except with the regressor equal to $\mathtt{Sae}(\mathbf{x})$ and the target equal to $\|\mathtt{Sae}(\mathbf{x})\|$ :

$$\boldsymbol{a}^* = \underset{\boldsymbol{a} \in \mathbb{R}}{\arg\min} \left\| \boldsymbol{a}^T \cdot \mathtt{Sae}(\mathbf{x}) - \|\mathtt{Sae}(\mathbf{x})\|_2^2 \right\|_2 \tag{21}$$

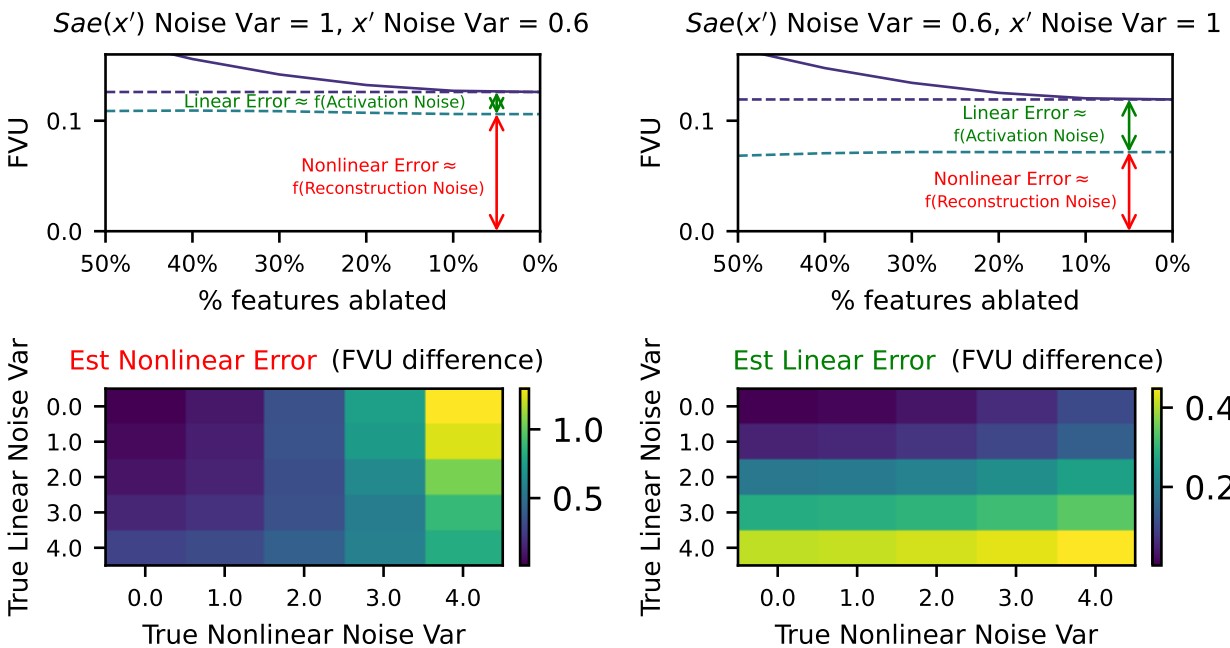

Figure 15: **Top:** When controlled amounts of noise are added to synthetic data `Sae(x')` and $\mathbf{x}'$, the result is a plot similar to Fig. 1. **Bottom:** The nonlinear and linear error estimates (as shown at top) accurately correlate with the amount of noise added. The exact correlation between synthetic added noise and resulting estimated error components across these noise levels are shown in Table 1

One important note is that we subtract the bias from `Sae(x)` so that it is purely a sparse sum of SAE features (this makes analysis easier). For each SAE latent from the SAE, we then compute

$$(\boldsymbol{a}^*)^T \cdot \texttt{latent}_i \tag{22}$$

Finally, we plot this dot product against the average latent activation in Fig. 14. If $\boldsymbol{a}^*$ indeed equals the sum of the latents weighted by their activation, as we predict in Appendix B.1, then these two quantities should be approximately equal, which we indeed see in the figure.

### C.3 Synthetic SAE Error Vector Experiments

The results for different Gaussian noise amounts versus percentage of features ablated are shown in Fig. 15. On this distribution of vectors, the test works as expected; the variance explained by `Sae(x)` $+ \boldsymbol{a}^T\mathbf{x}$ is a horizontal line proportional to `Introduced(x)`, while the gap between this horizontal line and the asymptote of the variance explained by `Sae(x)` is proportional to `Dense(x)`.

We also tried running this test on a sparse sum of *random* vectors, which did not work as well, possibly due to not including the structure of the SAE vectors (Giglemiani et al., 2024); see Appendix D for more details.

## D Synthetic Experiments with Random Data

For this set of experiments, we generated a random vector $\mathbf{x}'$ that was the sum of a power law of $100k$ random gaussian vectors in $\mathbb{R}^{4000}$ with expected $L_0$ of around 100. To simulate the SAE reconstruction and SAE error, we simply masked a portion of the vectors in the sum of $\mathbf{x}'$. Unlike the more realistic synthetic data case we describe in Appendix C.3, this did not work as expected: even in the case with no noise added to $\mathbf{x}'$ or the simulated reconstruction, the variance explained by the sum of the linear estimate of the error plus the reconstructed vectors plotted against the number of features "ablated" formed a parabola (with minimum variance explained in the middle region), as opposed to a straight line as in Fig. 15.

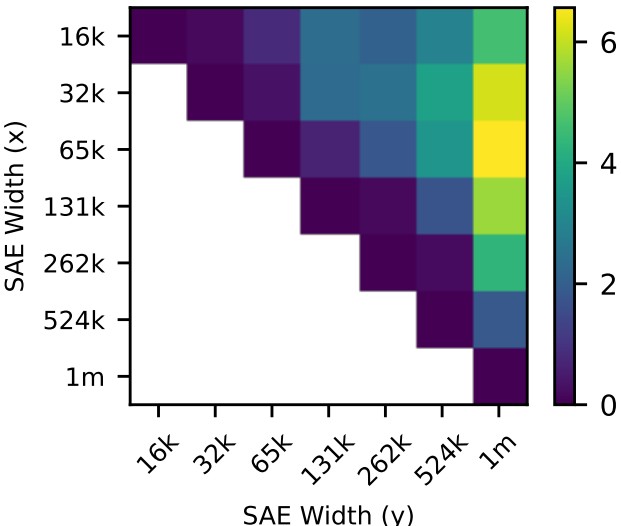

Figure 16: Percent decrease in FVU when additionally using the squared norms of nonlinear error to predict SAE error norm.

We note that this result is not entirely surprising: other works have found that random vectors are a bad synthetic test case for language model activations. For example, in the setting of model sensitivity to perturbations of activations, Giglemiani et al. (2024) found they needed to control for both sparsity and cosine similarity of SAE latents to produce synthetic vectors that mimic SAE latents when perturbed.

# E Using $\|\texttt{NonlinearError}(\mathbf{x})\|$ to Predict Scaling:

Following up on our discussion in Appendix B.3, we are interested in whether $\texttt{NonlinearError}(\mathbf{x})$ can help with predicting SAE per-token error norm scaling, as this might suggest that it contains a larger component of $\texttt{Introduced}(\mathbf{x})$ than $\texttt{LinearError}(\mathbf{x})$. Formally, we solve for

$$\boldsymbol{d}^* \coloneqq \arg\min_{\boldsymbol{d} \in \mathbb{R}^{\mathrel{\rlap{\hspace{0.1em}/}k}}} \left\| \boldsymbol{d}^T \cdot [\texttt{SaeError}_1(\mathbf{x}), \texttt{NonlinearError}_1(\mathbf{x})] - \texttt{SAE}_2(\mathbf{x}) \right\|_2 \tag{23}$$

To evaluate the improvement of $\boldsymbol{d}^*$ relative to $\boldsymbol{c}^*$ from Eq. (13), we report the percent decrease in FVU; see Fig. 16. We find that using the norm of $\texttt{NonlinearError}(\mathbf{x})$ provides a small but noticeable bump in the ability to predict larger SAE errors of up to a 5% decrease in FVU, validating this hypothesis.

