# OpenReview forum: "Decomposing The Dark Matter of Sparse Autoencoders"
_TMLR — Accepted by TMLR_

### Review · Reviewer_4PVn · 2024-12-28

**Summary Of Contributions:**

This paper investigates the sparse autoencoder (SAE) error of Gemma and shows that most of the error can be linearly predicted. Moreover, it empirically analyzes nonlinear SAE error and proposes two methods to reduce them.

**Audience:**

Yes

**Claims And Evidence:**

No

**Requested Changes:**

1. Preliminaries about sparse SAE would help readers better understand the research focus, particularly the SAE error vector.
2. This paper uses only Gemma Scope 9B model. However, it claims that "we are agnostic to the architecture or training procedure of the sparse autoencoder." To support this claim, additional experiments with other models should be included.
3. In Section 4, please explain why the 20th layer is chosen for experiments and provide further details on the specific activation locations used in these experiments.
4. The definition of $L0$ is not provided, yet it appears in several figures throughout the paper. A clear definition should be included.
5. Please clarify what the different lines in Figure 13 represent. Previous work (Scaling and Evaluating Sparse Autoencoders) suggests to use layers towards the end of the network. Does Figure 13 imply that the SAE error of these layers cannot be predicted linearly?

**Strengths And Weaknesses:**

- Strengths

  1. The paper is novel to analyze the SAE error and shows the interesting observation that SAE error vectors can be explained with a linear transformation.
  2. It provides a comprehensive validation of its hypotheses and conclusions on a large-scale language model, Gemma-2 9B.
  3. It proposes inference time gradient pursuit and linear transformations from earlier layer SAE outputs, which is useful and insightful to reduce nonlinear SAE error.

- Weaknesses

  1. This paper is difficult to understand. A major drawback of the paper is its complexity and lack of clarity in presentation. Some of the descriptions and technical details are ambiguous.
  2. The connection between explanations of SAE and interpretability of language models. The goal of SAE is to extract interpretable features or circuits from a language model, I do not clearly understand how the explanations in this paper can help to approach this goal.

---

> ### Author Response · Authors · 2025-02-05
>
> Thank you for your excellent comments and suggestions! We go through them here in detail. See also our global response above, where we summarize the changes we plan to make to the paper. Please let us know if anything is still unclear!
>
> >Preliminaries about sparse SAE would help readers better understand the research focus, particularly the SAE error vector.
>
> Thank you for this suggestion! We will add a discussion of preliminaries about SAEs to the next draft of the paper.
>
> > This paper uses only Gemma Scope 9B model. However, it claims that "we are agnostic to the architecture or training procedure of the sparse autoencoder." To support this claim, additional experiments with other models should be included.
>
> Thank you very much for this suggestion! As describe above in our global response on the changes that we intend to make, we plan to add experiments on Gemma Scope 2B, other layers of Gemma Scope 9B, and on Llama 3.1 8B.
>
> > In Section 4, please explain why the 20th layer is chosen for experiments and provide further details on the specific activation locations used in these experiments.
>
> We choose the 20th layer because it is a layer towards the end of the model (as you point out below, this was suggested in previous work) and because it has many SAEs in the Gemma Scope release. As discussed above, we now plan to run on more models and layers.
>
> > The definition of $L_0$ is not provided, yet it appears in several figures throughout the paper. A clear definition should be included.
>
> Thank you for this suggestion, we apologize this was unclear! We will add this definition in the revision of the paper.
>
> > Please clarify what the different lines in Figure 13 represent. Previous work (Scaling and Evaluating Sparse Autoencoders) suggests to use layers towards the end of the network. Does Figure 13 imply that the SAE error of these layers cannot be predicted linearly?
>
> Figure 13 shows that “simple” baselines (using model loss, L0, tokens, or act norm) do not work to predict SAE error norm in later layers while our method (using activations) does work. We will add more discussion of Figure 13, along with discussing this connection to previous work. Thank you for pointing this out!

---

### Review · Reviewer_8trg · 2025-01-22

**Summary Of Contributions:**

This paper studies the “dark matter” of Sparse Autoencoders (SAEs)  – the neural network activations that remain unexplained by the SAE decomposition. The authors begin by examining the SAE error using linear functions of the activations, and show that a significant portion of the error can be linearly predicted from the activations. To investigate the sources of the error, they decompose it into linear nonlinear components, and show that the nonlinear component is fundamentally different from the linear component. They further study what might constitute the linear and nonlinear components of the error using a model based on the weak linear representation hypothesis, and introduce a new term in the model corresponding to the error from learning the sparse features. Finally, the authors propose several strategies to reduce the nonlinear error and results suggesting that circuit analysis is likely relevant for studying SAE dark matter.

**Audience:**

Yes

**Broader Impact Concerns:**

No concerns.

**Claims And Evidence:**

No

**Requested Changes:**

Critical for acceptance:
1. Improve the clarity of the paper by addressing questions in the section above.
2. Include a wider range of layers and models in the experiments.

Will strengthen the paper:
1. Verify the almost-orthogonal assumption of the SAE features.

**Strengths And Weaknesses:**

Strengths:
1. The paper studies an important problem in mechanistic interpretability, namely the limitations of SAEs, and how to potentially overcome them.
2. The paper has interesting empirical insights, including the effectiveness of using linear predictors of the activations. It also includes experimental analyses from different perspectives, such as linear vs. nonlinear error and scaling.

Weaknesses:
1. Clarity: Sections 5 and 6 are difficult to understand. In particular, the definition of the error terms may contain typos, which make the results confusing:
    - Eq. (8): are the two terms in the introduced error flipped? If I flip the two terms, I get that SaeError(x)  = x - Sae(x).
    - What does the following sentence mean: “if this assumption is violated in practice, it will show up in our…”
    - In the second paragraph of Section 5.2, which test is referred to in “if this test is accurate”?
    - In the last paragraph of Section 5.2, which experiments show that Dense(x) and Introduced(x) are mostly constant?
    - Section 6, if I use the definition of LinearError in the definition of NonlinearError, I get: NonlinearError = SaeError - LinearError = SaeError - (SaeError - b* x) = b* x. This looks linear, is there a typo here?
    - Eq. (5), should there be norms surrounding SaeError_1(x) and SaeError_2(x), according to the exposition?
    - Why would Dense(x) be part of the linearly explainable error?
    - In Section 7.1, the authors state that having a better encoder doesn't necessarily reduce Introduced(x). Why would this happen empirically?

2. Limited experimental setup: although there are many kinds of analyses, almost all experiments are performed on a particular layer of a single model. Why was the model chosen, and why layer 20? How would the results change with the position of the layer and the model architecture? The paper will be much stronger if it includes more context around these choices and more diverse experimental setups. At the very least the paper should include discussions around the limitations of the current experimental setup.

3. Empirical verification of assumptions: In Section 5.1, the explanation builds on the intuition that the y_i’s are almost orthogonal. This seems to be a property that can be verified whether it holds in practice.

---

> ### Author Response · Authors · 2025-02-05
>
> Thank you for your excellent comments and suggestions! Below, we go through each one, please let us know if you have any additional comments. See also our global response above, where we summarize the changes we plan to make to the paper.
>
> > Why would Dense(x) be part of the linearly explainable error?
> In Section 7.1, the authors state that having a better encoder doesn't necessarily reduce Introduced(x). Why would this happen empirically?
> What does the following sentence mean: “if this assumption is violated in practice, it will show up in our…”
> In the second paragraph of Section 5.2, which test is referred to in “if this test is accurate”?
>
> Thank you for pointing out these problems with our breakdown of error into Dense and Introduced error, we apologize that we were not clearer! In the next version of our work we plan to move Section 5 to the appendix and present it as a potential hypothesis explaining our main body results, rather than driving assumptions. This will hopefully avoid many of these confusions. Then in the main body, we will only reference Sae(x), LinearError, and NonlinearError.
>
> > Section 6, if I use the definition of LinearError in the definition of NonlinearError, I get: NonlinearError = SaeError - LinearError = SaeError - (SaeError - b* x) = b* x. This looks linear, is there a typo here?
> Eq. (8): are the two terms in the introduced error flipped? If I flip the two terms, I get that SaeError(x) = x - Sae(x).
>
> This is our mistake, thank you for catching this! We plan to define the following terms in the main text:
> SaeError(x) = x - Sae(x)
> LinearError = b* x
> NonlinearError = SAEError - LinearError = Sae(x) - b* x
>
> > Eq. (5), should there be norms surrounding SaeError_1(x) and SaeError_2(x), according to the exposition?
>
> You are correct, we will fix that, thank you!
>
>
> > In the last paragraph of Section 5.2, which experiments show that Dense(x) and Introduced(x) are mostly constant?
>
> These are the experiments in Appendix B.3 on synthetic data that we link to; once we move this section to the appendix, hopefully this is more clear.
>
> > Limited experimental setup: although there are many kinds of analyses, almost all experiments are performed on a particular layer of a single model. Why was the model chosen, and why layer 20? How would the results change with the position of the layer and the model architecture? The paper will be much stronger if it includes more context around these choices and more diverse experimental setups. At the very least the paper should include discussions around the limitations of the current experimental setup.
>
> Thank you very much for this suggestion! As described above in our global response on the changes that we intend to make, we plan to add experiments on Gemma Scope 2B, other layers of Gemma Scope 9B, and on Llama 3.1 8B.
>
> > Empirical verification of assumptions: In Section 5.1, the explanation builds on the intuition that the y_i’s are almost orthogonal. This seems to be a property that can be verified whether it holds in practice.
>
> The problem here is that we do not have access to the unlearned features in the SAE error term, so we do not have the ability to tell if they are almost orthogonal. If we assume that the unlearned features roughly follow the distribution of existing SAE features, then we might be able to partially examine this; indeed, most learned SAE features are indeed almost orthogonal.

---

### Review · Reviewer_WABm · 2025-01-22

**Summary Of Contributions:**

This paper studies the “dark matter” of sparse autoencoders: the unexplained variance in the activations of SAEs when they are used to explain LLM layers. The authors perform an array of in-depth experiments showing many and interesting results, pertaining to the linear predictability of errors, nonlinear error analysis, insights about the scaling behavior, and postulating methods to reduce the SAEs error.

**Audience:**

Yes

**Broader Impact Concerns:**

No broader impact concerns.

**Claims And Evidence:**

Yes

**Requested Changes:**

**Requested Changes**

Overall I think the paper is in a good state with virtually no typos (the only one I could find was in page 7, next to Table 1, where “noise” is typed as “nosie”) and an interesting message. In terms of requested changes I would ask:

- Motivate (or emphasize) the bottom line: what are practical implications of the findings and how can they improve the models/understanding of other researchers?
- Attempt to restructure/streamline/explain the experimental section. As mentioned above, some sections are extremely hard to follow and the experiments feel disconnected: having multiple results is okay, however currently they are fairly confusing.
- Better address the related work, from the sparse coding perspective.

I also had some further comments on the experiments:

- In Figures 2, 3, and 4 the black dots are not explained. Are they the actual models and the rest are interpolations? If so, I believe their message is greatly hurt: the most interesting areas (where gradients are shown) are not evaluated and we don’t know if the trends are actually consistent and if the loss metrics are smoothly changing, which currently is a tenet of the work. Following up to this, why are areas with different point densities?
- In most experiments, SAEs are applied to layer 20 of Gemma. Why is this particular layer chosen? Are the results consistent across other layers and models? These questions seem very important to be able to generalize the findings of the work.
- As a slight follow-up to the above, part of Section 6 presents results on Llama: again, the choice of the architecture seems arbitrary and it’s not clear how one can compare results between different architectures. The impact of the experiment would be much stronger if it was on the same architecture.

**Strengths And Weaknesses:**

**Strengths**

Overall the paper is well-written and for the most part intuitive. To my knowledge the results are novel, and this is the first work I’ve seen studying the error of the SAEs. I think this can prove to be an interesting direction for future works, where further emphasis is placed on understanding the tools we use beyond performance metrics.

The paper also has a very extensive suite of experimental results, each interesting in its own right. I also particularly enjoy the allusion to traditional sparse coding.

**Weaknesses**

I think some of the strengths unfortunately are also weaknesses. Specifically I found some weaknesses around motivation, presentation and clarity, and related work.

*Motivation*

In terms of motivation, I personally struggled to find a very convincing argument as to why this is important: obviously the graphs have clear trends in them and there is a story to be told, however it is not clear to me how these results are actionable and can lead to better models or better understanding/interpretability of LLMs. It is possible that such a strong motivation was there and maybe I missed it, which brings me to the second point.

*Presentation*

There are a lot of results in the manuscript, which at times can feel disconnected. The reader has to invest a lot of energy in following all of the results and form a coherent story, and I think some restructuring and rewriting could probably help make the text more coherent. I think particularly problematic are sections 5.2 and onwards, where mentions are constantly made to “Introduced”, “Dense”, “LinearError”, etc, and it’s extremely difficult to keep track of all of these decompositions. A visual here would be very helpful, I believe, when a new decomposition is introduced, maybe also comparing it with the previous decomposition.

*Related work*

I was particularly excited to read about the adoption of the weak linear hypothesis, but then I found that there were nearly no references to traditional sparse coding, which has a very large and rich history of studying exactly the solutions of (6). Specifically, the model of (6), where a vector is decomposed in a sparse and dense component is very thoroughly studied in [1], including theory and applications to neural networks. Moreover, the original result of Section 5.1 is trivial in linear algebra/sparse coding but there are no references, and the “almost orthogonal” is needlessly vague. In sparse coding there are multiple ways to talk about this, from coherence, to the restricted isometry property, the nullspace property, and many more. I understand that most LLM papers these days do not engage with the sparse coding theory of SAEs, but without proper attribution we are doing a disservice to a longstanding and respected field, and at the same time run the risk of reinventing the wheel.

[1] Tasissa, Theodosis, Tolooshams, Ba, “Discriminative reconstruction via simultaneous dense and sparse coding”, TMLR 2024

---

> ### Author Response · Authors · 2025-02-05
>
> Thank you very much for you excellent comments and suggestions! Below, we respond to each of your points, please let us know if you have any more questions. We also have written up a global respond with changes we intend to make to the paper, and plan to upload a new version in around a week.
>
> > Overall I think the paper is in a good state with virtually no typos (the only one I could find was in page 7, next to Table 1, where “noise” is typed as “nosie”) and an interesting message.
>
> We are glad to hear that you think the paper is in a good state! We will fix that typo.
>
>
> > In terms of requested changes I would ask:
> Motivate (or emphasize) the bottom line: what are practical implications of the findings and how can they improve the models/understanding of other researchers?
>
> Thank you for this suggestion! We agree we could motivate the implications of our research more clearly, and have added a list of the implications we intend to discuss to our global comment above. Briefly, these implications are: to inspire new SAE architectures, make experiments with new SAE architectures more scalable, improve circuit analysis, and improve general scientific understanding of SAE errors.
>
> > Attempt to restructure/streamline/explain the experimental section. As mentioned above, some sections are extremely hard to follow and the experiments feel disconnected: having multiple results is okay, however currently they are fairly confusing.
>
> Thank you for pointing this out! We agree that Section 5 especially is confusing, and we also think that the error-breakdown into introduced, dense, and linear errors might not be needed to understand our experiments in later sections. Thus, we plan to streamline the paper by moving Section 5 to the appendix and reframing and reorganizing later experiments.
>
> > Better address the related work, from the sparse coding perspective.
>
> Thank you for pointing out this related work, we were not aware of it! We will add a discuss of this work, and a longer history of SAEs and sparse coding, to the related work section.
>
> > In Figures 2, 3, and 4 the black dots are not explained. Are they the actual models and the rest are interpolations? If so, I believe their message is greatly hurt: the most interesting areas (where gradients are shown) are not evaluated and we don’t know if the trends are actually consistent and if the loss metrics are smoothly changing, which currently is a tenet of the work. Following up to this, why are areas with different point densities?
>
> Yes, these are actual SAEs and the rest are interpolations; we will make this clearer in the paper. Because we are using pre trained Gemma Scope SAEs, we do not choose the locations of the points ourselves; part of the reason that we show layer 20 is because this is the Gemma Scope layer with the most SAEs available.
>
> We still believe that looking at the three horizontal and vertical lines of points in the plot (corresponding to groups of SAEs with the same L0 and same width) show the trends that we claim. We are unsure of a better way to present the data than a 2D histogram, but will experiment with multiple line plots during the next week.
>
> > In most experiments, SAEs are applied to layer 20 of Gemma. Why is this particular layer chosen? Are the results consistent across other layers and models? These questions seem very important to be able to generalize the findings of the work.
>
> This particular layer was chosen because there were the most SAEs available from Gemam Scope. We plan to add experiments on Gemma Scope 2B, other layers of Gemma Scope 9B, and on Llama 3.1 8B to Section 4.
>
> > As a slight follow-up to the above, part of Section 6 presents results on Llama: again, the choice of the architecture seems arbitrary and it’s not clear how one can compare results between different architectures. The impact of the experiment would be much stronger if it was on the same architecture.
>
> In section 6, we refer to Llama only in that we are using it as an explainer model to generate automated interpretability explanations of SAE features; we do not present any results on it. We apologize if we were not clear enough here, and will make this clearer in the next version of the paper!

---

### Author Response · Authors · 2025-02-05
**Summary of changes we plan to make**

We would again like to thank all of the reviewers for their excellent comments. We have now replied to all of the the reviewers concerns, and we will soon upload a new copy of the paper with the following changes:


**Clarity + Structure Updates:**
- We plan to move the theoretical discussion currently in Section 5 to the appendix, and present this discussion as a hypothesis rather than assumptions. We will then add additional experiments on other models and layers to Section 4. The new Section 5 will consist of what is currently Section 6 and will focus on experiments investigating the difference between the part of the SAE error component that is linearly predictable and the part that is not. We will frame this in a way that does not involve the conceptual definitions of Dense Error and Introduced Error that will now be in the appendix.
- We will add discussions of SAE basics, L0, and other necessary terminology to the notation section.
- We will add a discussion on classical sparse coding techniques to the related work.

**More models and layers:**
- We will run on Llama Scope SAEs, which vary in layer and width.
- We will run on Gemma Scope 9B and additional layers on 2B.
- We will add these plots to section 4.

**Motivation/implications**
We plan to add discussion surrounding the motivation and implications of our work to the introduction and conclusion. Specifically, we plan to discuss the following potential impacts of our work:
- *Inspire new SAE architectures*: we may want to design new SAE architectures that reduce the component of error that can be predicted from activations, as these components seem to represent persistently “unlearned” features. For instance, we might try adding a low rank dense side channel to the SAE.
- *Make experiments with new SAE architectures more scalable*: since we show that one can predict the error in a larger SAE by using a small SAE, we might be able to efficiently predict how new SAE architectures will scale by examining their per-token SAE error behavior on small SAEs.
- *Improve circuit analysis*: We show that one can decrease the magnitude of an SAE’s error by using past SAE outputs, which might help reduce “noise terms” in SAE circuit analysis.
- *General scientific understanding*: Our work is the first work breaking down SAE errors, and we hope that it might overall give more intuition on the nature of errors that SAEs make.

We will make another comment when this new version is uploaded, most likely in about a week. Please let us know if you have any other questions in response to our new comments.

---

### Author Response · Authors · 2025-02-06
**New draft with changes highlighted**

We have now submitted a new draft of our paper that contains the fixes and improvements suggested by the reviewers, with all important changes highlighted in **blue**. Thank you again to all of the reviewers for their extremely helpful feedback; we believe that after making these changes the the paper is significantly stronger and clearer.

---

### Decision · Action_Editor_KXY2 · 2025-03-19

**Recommendation:** Accept as is

**Comment:**

Overall, the paper is well structured, studies an interesting and important problem, and presents a compelling message.

However, as the reviewers have pointed out, the evidence is a bit fuzzy. The experiments about predicting the linear term in the error seem to indicate a stable finding; however, the prediction of the error norms seem to hold only for the middle layers, and there seem to be no special insights about the nonlinear terms. Finally, it is unclear (at least at this point) how exactly the proposed decomposition, of SAE dark matter into finer-grained error terms, contributes to *actionable* insights towards improving SAEs or other circuit interpretation methods.

Overall, I recommend an accept since but encourage the authors to consider these points (and the reviewers' feedback) while pursuing this line of work further.

**Audience:**

The paper analyzes a popular technique in mechanistic interpretability of language models which is definitely of interest to (a subset of) the TMLR audience.

**Claims And Evidence:**

The paper presents an interesting set of findings on the efficiency of sparse autoencoders (SAE) for representing language model activations, measured in terms of the fraction of unexplained variance (or "dark matter"). The hypothesis is that this error term can be further broken down into an extra linear term and a residual nonlinear term, and evidence (based on experiments on Gemma 9B and Llama 3.1 8B) is presented to support this hypothesis. This findings suggest that current techniques current SAE do not capture a linear (dense) subspace in the activation space, giving indications that techniques other than scaling up SAEs may be necessary to explain more of model behavior.